# RAD: Retrieval High-quality Demonstrations to Enhance Decision-making

**Lu Guo**[1]  **Yixiang Shan**[1]  **Zhengbang Zhu**[2]  **Qifan Liang**[1]  **Lichang Song**[1]
**Ting Long**[1]  **Weinan Zhang**[2]  **Yi Chang**[1]

## Abstract

Offline reinforcement learning (RL) learns policies from fixed datasets, thereby avoiding costly or unsafe environment interactions. However, its reliance on finite static datasets inherently restricts the ability to generalize beyond the training distribution. Prior solutions based on synthetic data augmentation often fail to generalize to unseen scenarios in the (augmented) dataset. To address these challenges, we propose Retrieval High-quAlity Demonstrations (RAD) for decision-making, which innovatively introduces a retrieval mechanism into offline RL. Specifically, RAD retrieves high-return and reachable states from the offline dataset as target states, and leverages a generative model to generate subtrajectories conditioned on these targets for planning. Since the targets are high-return states, once the agent reaches such a target, it can continue to obtain high returns by following the associated high-return actions, thereby improving policy generalization. Extensive experiments confirm that RAD achieves competitive or superior performance compared to baselines across diverse benchmarks, validating its effectiveness. Our code is available at https://github.com/LeahGL/RAD.

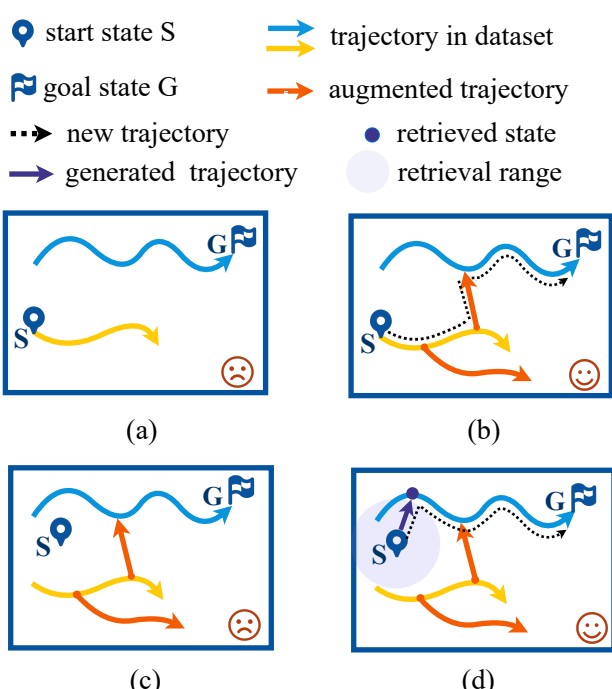

*Figure 1.* (a) Far-apart trajectories in the offline dataset make it difficult for the policy to learn how to reach $G$ from $S$. (b) By connecting segments, augmentation-based methods expand the dataset and facilitate learning a policy that can reach the $G$ from $S$. (c) When the initial state $S$ falls into an out-of-distribution (OOD) region again, the old augmented data cannot support the agent in learning a policy from $S$ to $G$. (d) RAD dynamically retrieves high-value and reachable states as intermediate targets to guide the agent from $S$ to $G$.

## 1. Introduction

Offline reinforcement learning (RL) aims to learn effective decision policies purely from static datasets, without further interaction with the environment (Levine et al., 2020; Prudencio et al., 2023; Park et al., 2024). This setting is essential for domains where active exploration is costly or unsafe, such as robotics (Kalashnikov et al., 2021), health-

care (Fatemi et al., 2022), and autonomous driving (Shi et al., 2021). Despite promising advances, offline RL faces a fundamental limitation: the finite scale of static datasets inherently restricts the learned policy's ability to generalize beyond the training distribution. As illustrated in Figure 1(a), it is challenging to learn a policy that enables the agent to reach the target state $G$ from the initial state $S$ using an offline dataset containing only two trajectories. This is because the two trajectories are too far apart, making it difficult for existing offline RL algorithms to generalize to the transition from $S$ to $G$.

To overcome this, recent works typically generate transi-

---

[1]School of Artificial Intelligence, Jilin University, Changchun, China [2]School of Computer Science, Shanghai Jiao Tong University, Shanghai, China. Correspondence to: Ting Long <tinglong@jlu.edu.cn>, Yi Chang <yichang@@jlu.edu.cn>.

*Proceedings of the 43rd International Conference on Machine Learning*, Seoul, South Korea. PMLR 306, 2026. Copyright 2026 by the author(s).

tions to augment the original dataset, alleviating the negative impact of finite static datasets in offline setting (Lu et al., 2023; Li et al., 2024). As shown in Figure 1(b), a new sub-trajectory is generated by the augmentation-based methods, which enable the learning of the policy to support the agent to start from state $S$ and reach state $G$. However, these augmentations are typically generated in a static offline manner, which lack flexibility. Once generated, the augmented dataset remains fixed and cannot adapt to dynamic situations, as shown in Figure 1(c): if the agent later encounters a new state (e.g., a different start state out of the distribution of augmentation and original dataset), there may be no existing demonstration or augmented path that provides meaningful guidance. Consequently, the policy may fail to generalize again, especially under distributional shifts or changing task demands. This highlights the brittleness and limited flexibility of static augmentation methods in offline RL.

A promising approach to achieve effective generalization in offline RL is to adopt an adaptive mechanism: one that adaptively stitches to high-reward trajectories within a certain range to escape out-of-distribution situations or low-reward scenarios. As it is illustrated in Figure 1(d), starting from a new state $S$, the agent first plans towards a state located along a high-return trajectory. Once the agent reaches this state, it can then easily navigate to the target state $G$ by leveraging the experience from the high-return trajectory. Inspired by that, we propose Retrieval High-quAlity Demonstrations (RAD) in this paper, which is built upon a retrieval mechanism and a generative model. It uses the retrieval mechanism to select states from high-return trajectories in the surrounding region as target states for planning. The agent then leverages the generative model to generate subsequent trajectories toward the target state to achieve higher rewards through interaction with the environment. In such a manner, RAD can efficiently facilitate the transition from OOD states or low-return regions to potentially high-return states without relying on complex data augmentation processes. We conduct extensive experiments on D4RL dataset, and the experiment results demonstrate the effectiveness of the RAD.

Our main contributions are: (i) We propose RAD, which retrieves states from high-return trajectories as the target for planning; (ii) RAD can efficiently facilitate the transition from OOD states or low-return regions to potentially high-return states without relying on complex data augmentation processes; (iii) The extensive experiments demonstrate the superiority of RAD.

## 2. Related Work

Offline reinforcement learning (RL) aims to learn policies from static datasets without additional environment interac-

tion (Levine et al., 2020; Prudencio et al., 2023). The most straightforward solution is behavior cloning (BC), which treats offline RL as a supervised learning problem by directly imitating the behavior policy in the dataset. Another line of work reformulates policy learning as a sequential modeling problem (Chen et al., 2021; Janner et al., 2021). For example, Decision Transformer (DT) (Chen et al., 2021) conditions on the return-to-go and models entire trajectories with a Transformer, enabling long-horizon planning. More recently, diffusion-based methods (Ajay et al., 2022; Janner et al., 2022; Dong et al., 2024) such as Diffuser (Janner et al., 2022) apply generative diffusion models to synthesize trajectories, showing strong performance across various offline RL benchmarks. Despite these advances, most of these methods struggle to generalize beyond the distribution of the offline dataset. Conservatism-based approaches (Kumar et al., 2020; Yu et al., 2020; Kidambi et al., 2020), such as CQL (Kumar et al., 2020) and MOPO (Yu et al., 2020), attempt to mitigate extrapolation errors by constraining the learned policy within the support of the dataset, either by penalizing out-of-distribution actions or introducing uncertainty-aware rollouts. However, these methods fundamentally keep the policy restricted to the offline dataset distribution and cannot fully exploit potentially better behaviors outside it. Data augmentation approaches (Lu et al., 2023; Li et al., 2024), such as Synthetic Experience Replay (SER) (Lu et al., 2023) and DiffStitch (Li et al., 2024), enrich the dataset by generating or stitching trajectories, partially alleviating OOD issues. Yet, once trained on the augmented dataset, the policy often fails to adapt to new states beyond the synthesized distribution.

To address the issue, we propose a method called Retrieval High-quAlity Demonstrations (RAD), which retrieves high-return states from the surrounding region and plans toward them to better handle decision-making under OOD conditions, thereby guiding the policy to achieve higher rewards.

## 3. Preliminary

### 3.1. Diffusion Model

Diffusion Models (Sohl-Dickstein et al., 2015; Song et al., 2020; Ho et al., 2020) are the generative models, which typically have two processes: a forward process and a reverse process. In the forward process, given a clean sample $\boldsymbol{x} \sim q(\boldsymbol{x})$, diffusion models treat $\boldsymbol{x}$ as the initial sample $\boldsymbol{x}^0$, and inject Gaussian noise step by step with $q(\boldsymbol{x}_t \mid \boldsymbol{x}_{t-1}) = \mathcal{N}(\boldsymbol{x}_t \mid \sqrt{1 - \beta_t}\,\boldsymbol{x}_{t-1}, \beta_t \boldsymbol{I})$, where $\boldsymbol{I}$ is the identity matrix, and $\beta_t$ controls the noise level at step $t$. As the forwarding process progresses, the sample becomes increasingly corrupted by noise. After $K$ steps, sample $\boldsymbol{x}$ is transformed into pure Gaussian noise $\boldsymbol{x}^K$. The reverse process starts from a pure Gaussian noise, it aims to recover $\boldsymbol{x}$ by gradually removing the noise step by step

with $p_\theta(\boldsymbol{x}_{t-1} \mid \boldsymbol{x}_t) = \mathcal{N}(\boldsymbol{x}_{t-1} \mid \mu_\theta(\boldsymbol{x}_t, t), \Sigma_\theta(\boldsymbol{x}_t, t))$, where the mean can be re-expressed with $\mu_\theta(\boldsymbol{x}_t, t) = \frac{\sqrt{\alpha_t}(1-\bar{\alpha}_t)}{1-\bar{\alpha}_{t-1}} \boldsymbol{x}_t + \frac{\sqrt{\bar{\alpha}_{t-1}}\beta_t}{1-\bar{\alpha}_t} \phi_\theta(\boldsymbol{x}_t, t)$, with $\alpha_t = 1 - \beta_t$ and $\bar{\alpha}_t = \prod_{s=1}^{t} \alpha_s$, $\phi_\theta$ is model to reconstruct $\boldsymbol{x}$. Fixing $\Sigma_\theta(\boldsymbol{x}_t, t) = \beta_t I$ (Ho et al., 2020), the learning objective is formulated by minimizing the mean squared error between the true signal and the model prediction:

$$\mathcal{L} = \mathbb{E}_{\boldsymbol{x}, t \sim [1,T]} \left[ \| \boldsymbol{x}^0 - \psi_\theta(\boldsymbol{x}_t, t) \|^2 \right]. \tag{1}$$

### 3.2. Problem Definition

RL is typically formulated as a Markov Decision Process (MDP). Formally, a MDP is given by $\mathcal{M} = \{\boldsymbol{S}, \boldsymbol{A}, P, r, \gamma\}$, where $\boldsymbol{S}$ is the state space, $\boldsymbol{A}$ is the action space, $P$ is the transition function, $r$ is the reward function, and $\gamma \in (0, 1)$ is the discount factor. At each timestep $t$, the agent observes the environment state $\boldsymbol{s}_t$, takes an action $\boldsymbol{a}_t$ according to a policy $\pi_\theta$ parameterized by $\theta$, then receives an instantaneous reward $r_t$ from environment, and transits to state $\boldsymbol{s}_{t+1}$ via $P(\boldsymbol{s}_{t+1} \mid \boldsymbol{s}_t, \boldsymbol{a}_t)$. The interaction history is represented as a trajectory $\tau = \{(\boldsymbol{s}_t, \boldsymbol{a}_t, r_t) \mid t \geq 0\}$. We define the cumulative discounted reward from step $t$ as $v_t = \sum_{i>t} \gamma^{i-t} r_i$, and refer to it as the return of state $\boldsymbol{s}_t$. Additionally, the return of a complete trajectory $\tau$ is defined as $R(\tau) = \sum_{t \geq 0} \gamma^t r_t$.

We focus on the offline RL setting, where the agent cannot interact with the environment and must learn from a fixed dataset $\mathcal{D} = \{\tau_i\}_{i=1}^{N}$ consisting of $N$ trajectories collected by some unknown behavior policy. Each trajectory $\tau_i$ is a sequence of state-action-reward tuples: $\tau_i = \{(\boldsymbol{s_0}, \boldsymbol{a_0}, r_0), (\boldsymbol{s_1}, \boldsymbol{a_1}, r_1), \ldots, (\boldsymbol{s_{T-1}}, \boldsymbol{a_{T-1}}, r_{T-1})\}$, where $T$ denotes the length of each trajectory. Our goal is to learn a policy $\pi_\theta$ that maximizes the expected return without interacting with the environment:

$$\pi_\theta = \arg\max_\theta \mathbb{E}_{\tau \sim \pi_\theta}[R(\tau)]. \tag{2}$$

## 4. Method

We propose a method called Retrieval High-quAlity Demonstrations (RAD) for offline RL, which integrates a retrieval augmented mechanism with sub-trajectory generation to improve policy generalization under the scenarios beyond the dataset coverage. As it is illustrated in Figure 2, RAD is composed of a target selection (TS) module, a step estimation (SE) module, and a planning (PL) module. Given the current state $\boldsymbol{s}_t$, TS first retrieves reachable and high-return states as targets as $\boldsymbol{s}_t^g$. SE then estimates the step $\hat{i}_t$ transit from the current state $\boldsymbol{s}_t$ to the target state $\boldsymbol{s}_t^g$. PL finally randomly initializes a noisy sub-trajectory, and offsets $\boldsymbol{s}_t$ and $\boldsymbol{s}_t^g$ to the first position and the $\hat{i}_t$-th position, generating the subsequent trajectory and making a decision. Since the targets are high-return states, once the agent reaches

such a target state, it can obtain a high return by following the high-return action associated with the target state, thereby addressing the generalization of policy in low-return or OOD regions. In the following, we will discuss the TS and SE first, and subsequently the PL.

### 4.1. Target Selection(TS) Module

The target selection (TS) module aims to dynamically select a high-return and reachable target state $\boldsymbol{s}_t^g$ from the offline dataset for the subsequent planning. To conduct that, we first construct a database that contains the states from expert trajectories (please refer to the Appendix C for more details). Each entry in the database is composed of: (1) state $\boldsymbol{s}_i$: the feature vector representing the environment state at timestep $i$; (2) trajectory ID: the identifier that indicates which trajectory the state $\boldsymbol{s}_i$ belongs; (3) timestep $i$: the time step of state $\boldsymbol{s}_i$ within its trajectory; (4) discounted return $v_i$ : the cumulative discounted return starting from state $\boldsymbol{s}_i$.

Based on the current state $\boldsymbol{s_t}$ and the database, we take the following steps to obtain the target state:

**Selecting the similar states:** Given the current state $\boldsymbol{s_t}$, we first use it as a query vector to retrieve states from the database based on their similarity to $\boldsymbol{s_t}$. We employ two metrics to measure state similarity. For locomotion and manipulation tasks, the similarity between the current state $\boldsymbol{s_t}$ and database states is computed using the cosine similarity. For navigation tasks, the similarity is measured by the Euclidean distance in the two-dimensional spatial plane. We then select the top-$k$ states with similarity greater than $\delta$ and include them in the set $\mathcal{C}_s$.

**Extracting the high-return states:** For each state $\boldsymbol{s_i} \in \mathcal{C}_s$, we consider the subsequent trajectory within the next $H-1$ steps to ensure a sufficient number of candidate states, and compute the cumulative discounted return starting from each state. Specifically, for any state $\boldsymbol{s_j}$ with $i \leq j \leq H-1$ in the subsequent trajectory of $\boldsymbol{s_i}$, its cumulative discounted return is denoted as $v_j$. We then extract those states whose return lies within a tolerance threshold $\eta$ of the highest return $v^*$ :

$$|v_j - v^*| \leq \eta, \tag{3}$$

where $v^*$ denotes the maximum return among the candidate states in the batch. The states satisfying this condition are collected into the set $\mathcal{C}_g$.

**Drawing out the trajectory-continuable state:** To improve planning robustness and provide a richer context for the trajectory generation module, we prioritize candidates associated with longer remaining trajectories in the original demonstrations. Concretely, let $\ell_j$ denote the length of the remaining trajectory starting from $\boldsymbol{s}_j, \boldsymbol{s}_j \in \mathcal{C}_g$, we select

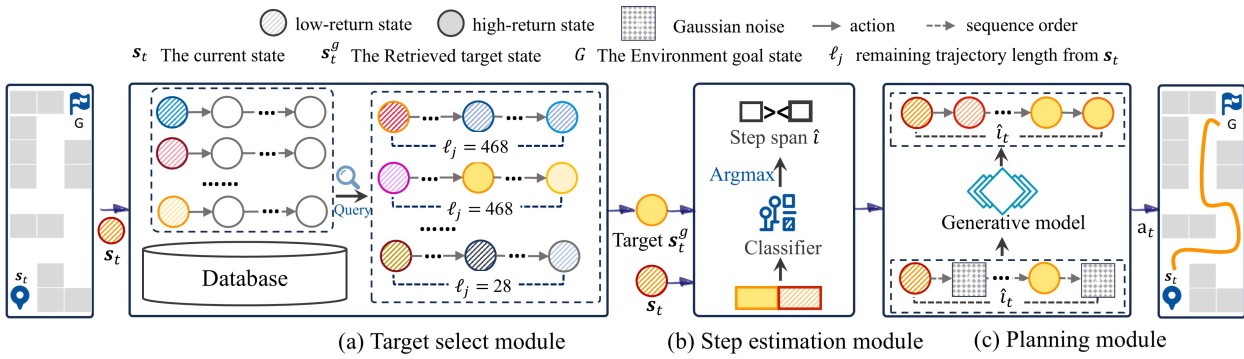

*Figure 2.* Overall framework of RAD.

the target state with:

$$s_t^g = \arg\max_{s_j \in \mathcal{C}_g} \ell_j. \tag{4}$$

The selected target state $s_t^g$ is then used to guide downstream planning.

### 4.2. Step Estimation(SE) Model

While the retrieved target state $s_t^g$ specifies the direction the agent should move toward for high return, but does not indicate the temporal distance, *i.e.*, the number of steps needed to reach the target state from the current state, making it difficult for the agent to plan effectively. From the perspective of Markov Decision Processes and conditional generative modeling, capturing temporal alignment is crucial (Ajay et al., 2022): without modeling the expected arrival step, the generated sub-trajectories are prone to overshooting, stalling, or degenerating, leading to incoherent or infeasible behaviors. Therefore, we design the SE module to predict the step span between the current state $s_t$ and the target state $s_t^g$. However, directly predicting the step span is a discrete regression problem, which is difficult than classification (Xiong & Yao, 2022). To alleviate this difficulty, we reformulate the regression problem as a classification task. Specifically, we concatenate the current state $s_t$ with the target state $s_t^g$, and feed the concatenation to a multilayer perceptron $f_e$:

$$e_t = \text{Sigmoid}(f_e([s_t, s_t^g])), \tag{5}$$

where $[.,.]$ denotes the concatenation operation, and $e_t$ is a $H-1$-dimensional vector, the i-th dimension represents the probability of $s_t$ requires $i$ steps to reach $s_t^g$. Then, we obtain the estimated step span with :

$$\hat{i} = \arg\max e_t. \tag{6}$$

### 4.3. Planning(PL) Module

With the current state $s_t$, target state $s_t^g$ and the estimated step count $\hat{i}$, the PL module aims to apply a diffusion model

to generate the subsequent trajectory for planning. Specifically, we randomly initialized a noisy sub-trajectory $\tau_{temp}$ with length of $H(H \geq \hat{i})$:

$$\tau_{temp} = \{\hat{\psi}_t^K, \hat{\psi}_{t+1}^K, \ldots, \hat{\psi}_{t+H}^K\}, \tag{7}$$

where each element $\hat{\psi}_t^K$ in $\tau_{temp}$ represents either a noisy state-action pair ($\hat{\psi}_t^K = \{\hat{s}_t^K, \hat{a}_t^K\}$) or a noisy state ($\hat{\psi}_t^K = \hat{s}_t^K$) only, and $K$ denotes the diffusion steps. Then, we obtain $\hat{\tau}_t^K$ from $\tau_{temp}$ by substituting the state in $\hat{\psi}_t^K$ with the current state $s_t$, and the state in $\hat{\psi}_{t+\hat{i}}^K$ with the target state $s_t^g$.

Starting from $\hat{\tau}_t^K$, we conduct the reverse denoising process of the diffusion model (Janner et al., 2022) to obtain a clean sub-trajectory. Each denoising step is parameterized as:

$$p_\theta(\hat{\tau}_t^{k-1}|\hat{\tau}_t^k) = \mathcal{N}(\mu_\theta(\hat{\tau}_t^k, k) + \rho\nabla\mathcal{J}_\phi(\hat{\tau}_t^k), \beta_k \boldsymbol{I}), \tag{8}$$

$$\mu_\theta(\hat{\tau}_t^k, k) = \frac{\sqrt{\alpha^k}(1-\bar{\alpha}^{k-1})}{1-\bar{\alpha}^{k-1}}\hat{\tau}_t^i + \frac{\sqrt{\bar{\alpha}^{k-1}}\beta^k}{1-\bar{\alpha}^k}\hat{\tau}_t^{k,0}. \tag{9}$$

Here $\hat{\tau}_t^{k,0} = \phi_\theta(\hat{\tau}_t^k, k)$ represents the $\tau_t^0$ constructed from $\hat{\tau}_t^k$ at diffusion step $k$, $\phi_\theta(\cdot, \cdot)$ is a network for trajectory generation, $k \sim [1, K]$ is the diffusion step, $\rho$ is a scaling factor controlling the guidance strength, $\mathcal{J}_\phi(\cdot)$ predicts the trajectory return to provide guidance for generation. $I$ denotes the identity matrix, and $\beta_i$ is the noise schedule coefficient that determines the proportion of noise injected at denoising step $i$. After $K$ denoising steps, we obtain the generated sub-trajectory $\hat{\tau}_t^0 = \{\hat{\psi}_t^0, \hat{\psi}_{t+1}^0, \ldots, \hat{\psi}_{t+H}^0\}$.

If the element in $\tau_{temp}$ is composed of a noisy state-action pair, $\hat{\tau}_t^0$ is the sequence of clean state-action pairs, we directly take the action in $\hat{\psi}_{t+1}^0$ to interact with the environment. If the element in $\tau_{temp}$ is composed of noisy states only, $\hat{\tau}_t^0$ is the sequence of clean states, and we then take out the state in $\hat{\psi}_{t+1}^0$, and feed it with the current state $s_t$ to a inverse dynamic model to obtain the action to interact with the environment:

$$a_t = f_a(s_t, \hat{s}_{t+1}), \tag{10}$$

where $\hat{s}_{t+1}$ denotes the generated states for step $t + 1$, $f_a$ is the inverse dynamic model.

Considering existing diffusion-based offline RL methods have already demonstrated strong quality in generating subsequent trajectories, and we focus on improving offline RL through a retrieval-augmented mechanism rather than trajectory generation itself, we select Diffuser (Janner et al., 2022) and DiffuserLite (Dong et al., 2024), two diffusion-based but totally different methods[1], to conduct the generation in implementation Eq. (8). Correspondingly, we have two variants: (1) **D-RAD**, which integrates our retrieval-augmented mechanism with the trajectory generation of Diffuser, producing trajectories of states and actions for decision making; (2) **DL-RAD**, which integrates our retrieval-augmented mechanism with the trajectory generation of DiffuserLite, producing trajectories of states only, after which actions are predicted using an inverse dynamics model for decision making.

### 4.4. Model Learning

Our method is trained with three losses: (1) the generation loss, which constrains the generation of planning toward high-return states; (2) the generation guidance loss, which constrains the guidance function; (3) the step estimation loss, which guarantees the accuracy of step estimation.

**Generation loss.** To train the generation of planning toward high-return states, we employ a pseudo target strategy. Concretely, we first sample a demonstration trajectory from the offline dataset:

$$\tau_t^0 = \{\psi_t^K, \psi_{t+1}^K, \ldots, \psi_{t+H-1}^K\}, \qquad (11)$$

where $\psi_t^K$ denotes the vector representation of the state (the state-action pair if the planning module is used to generate the state-action pair) of step $t$. Then, we randomly select an offset $i \sim \mathcal{U}(1, H-1)$ and set the vectors in $\psi_{t+i}^K$ as the pseudo target and applying forward diffusion to the sub-trajectory, the denoising network $\phi_\theta$ is trained by minimizing the noise prediction error:

$$\mathcal{L}_d = \mathbb{E}_{\boldsymbol{\tau}_t \in \mathcal{D}, t>0, k \sim [1,K]} \left[ \|\boldsymbol{\tau}_t - \phi_\theta(\hat{\boldsymbol{\tau}}_t^k, k)\|^2 \right], \quad (12)$$

**Generation guidance loss.** The generation guidance $\mathcal{J}_\phi(\cdot)$ is optimized by minimizing the mean squared error between the predicted trajectory return signal and the ground-truth return signal over the offline dataset $\mathcal{D}$:

$$\mathcal{L}_g = \mathbb{E}_{\tau \sim \mathcal{D}} \left[ (\mathcal{J}_\phi(\tau) - C(\tau))^2 \right]. \qquad (13)$$

---

[1]Our retrieval-augmented mechanism is, in theory, compatible with any trajectory–generation–based offline RL algorithm.

For D-RAD, $C(\tau)$ corresponds to the cumulative discounted return of the trajectory $R(\tau)$. For DL-RAD, $C(\tau) = \sum_{t=0}^{H-2} \gamma^t r_t + \gamma^{H-1} V(\boldsymbol{s}_{H-1})$, where $V(\boldsymbol{s}_t) = \max \mathbb{E}_\pi \left[ \sum_{\tau=t}^{\infty} \gamma^{\tau-t} r_\tau \right]$ denotes the optimal value function and can be estimated by a neural network through various offline RL methods. Here, $H$ is the temporal horizon.

**Step estimation loss.** The step estimation loss is formulated as the cross-entropy loss between the predicted step distribution $\boldsymbol{e}_t$ and the ground-truth step count $i$:

$$\mathcal{L}_e = -\mathbb{E}_{(\boldsymbol{s}_t, \boldsymbol{s}_t^g)} \left[ \log \boldsymbol{e}_t[i] \right], \qquad (14)$$

where $i$ is the ground truth offset of steps from $\boldsymbol{s}_t$ to $\boldsymbol{s}_t^g$, and $\boldsymbol{e}_t[i]$ denotes the predicted probability for class $i$.

$\mathcal{L}_d$, $\mathcal{L}_g$ and $\mathcal{L}_e$ are optimized independently. The details of the training and testing process are presented in Appendix B.

## 5. Experiment Design and Results Analysis

We explored the performance of RAD on a variety of offline RL tasks to answer the following research questions (RQs): (1) How does RAD perform compared with baseline methods across different environments? (2) Can RAD generalize to new states not covered in the training dataset? (3) How does the key component contribute to the performance of RAD? (4) Are the target states generated by RAD feasible and achievable in practice, and do they provide effective guidance for reaching the final goal? (5) What is the computational cost of RAD during training and per-step inference, compared with baseline methods?

### 5.1. Experiment Settings

**Environments and Datasets.** We evaluate the algorithm on various offline RL environments, including locomotion in Gym-MuJoCo (Brockman et al., 2016), long-horizon navigation in AntMaze (Fu et al., 2020), real-world manipulation in FrankaKitchen (Gupta et al., 2019), and 2D navigation tasks in Maze2D (Fu et al., 2020). Additional evaluations on OGBench (Park et al., 2025) are provided in Appendix D.10. We train models using publicly available datasets (see appendix A for further details).

**Baselines.** To evaluate our RAD, we compare it against a diverse representative offline RL algorithms. These include imitation learning methods such as Behavior Cloning (BC); model-free offline reinforcement learning approaches, including Conservative Q-Learning (CQL) (Kumar et al., 2020) and Implicit Q-Learning (IQL) (Kostrikov et al., 2021); model-based methods such as Model-based Offline Policy Optimization (MOPO) (Yu et al., 2020) and Model-based Offline Reinforcement Learning (MOReL) Kidambi et al. (2020); return-conditioned methods such as Decision

*Table 1.* The Performance across benchmark environments[2]. The results correspond to the mean over 3 random seeds with standard errors. Scores within 5% of the maximum per task ($\geq 0.95 \times$ MAX) are highlighted in **bold**. We abbreviate Diffuser as Diff and DiffuserLite as Lite for brevity.

| Dataset | Env | BC | CQL | IQL | MOPO | MOReL | DT | SER | DStitch | DS | DD | Diff | Lite | D-RAD | DL-RAD |
|---|---|---|---|---|---|---|---|---|---|---|---|---|---|---|---|
| Medium-Expert | HalfCheetah | 55.2 | **91.6** | 86.7 | 63.3 | 53.3 | 86.8 | 88.9 | **94.4** | **95.7** | 90.6 | 88.9 | 88.5 | 84.9 ± 0.5 | 90.1 ±0.1 |
| | Hopper | 52.5 | 105.4 | 91.5 | 23.7 | **108.7** | **107.6** | 110.4 | 110.9 | 107.0 | **111.8** | 103.3 | **111.6** | **112.3 ± 0.3** | 110.0 ± 0.3 |
| | Walker2d | **107.5** | 108.8 | 109.6 | 44.6 | 95.6 | 108.1 | 111.7 | 111.6 | 108.0 | 108.8 | 106.9 | 107.1 | 108.1 ±0.1 | 110.2 ±0.2 |
| Medium | HalfCheetah | 42.6 | 44.0 | **47.4** | 42.3 | 42.1 | 42.6 | **49.3** | **49.4** | 47.8 | **49.1** | 42.8 | 48.9 | 44.2 ±0.2 | **48.8 ±0.6** |
| | Hopper | 52.9 | 58.5 | 66.3 | 28.0 | 95.4 | 67.6 | 66.6 | 71.0 | 76.6 | 79.3 | 74.3 | 100.9 | 82.5 ±2.3 | **101.0 ±1.1** |
| | Walker2d | 75.3 | 72.5 | 78.3 | 17.8 | 77.8 | 74.0 | **85.9** | 83.2 | 83.6 | 82.5 | 79.6 | 88.8 | 82.8 ±0.7 | 89.4 ±0.2 |
| Medium-Replay | HalfCheetah | 36.6 | 45.5 | 44.2 | **53.1** | 40.2 | 36.6 | 46.6 | 44.7 | 41.0 | 39.3 | 37.7 | 41.6 | 41.2 ±0.1 | 44.4 ±0.1 |
| | Hopper | 18.1 | 95.0 | 94.7 | 67.5 | 93.6 | 82.7 | **102.4** | **102.1** | 89.5 | 100.0 | 93.6 | 96.6 | **98.0 ±0.6** | 100.4 ±0.4 |
| | Walker2d | 26.0 | 77.2 | 73.9 | 39.0 | 49.8 | 66.6 | 85.7 | 86.6 | 80.7 | 75.0 | 70.6 | 90.2 | 77.6 ±1.2 | 93.5 ±1.2 |
| Average | | 51.9 | 77.6 | 77.0 | 42.1 | 72.9 | 74.7 | 83.1 | **83.8** | 81.1 | 81.8 | 77.5 | **86.0** | 81.3 | **87.5** |
| Play | Antmaze-Medium | 0.0 | 65.8 | 65.8 | 0.0 | 0.0 | 0.0 | 41.0 | 65.8 | 0.0 | 8.0 | 6.7 | 78.0 | 40.0 ± 5.2 | **86.7± 3.6** |
| | Antmaze-Large | 0.0 | 20.8 | 42.0 | 0.0 | 0.0 | 0.0 | 72.9 | 42.0 | 0.0 | 0.0 | 17.3 | 72.0 | 13.3 ± 3.6 | **80.0± 4.2** |
| Diverse | Antmaze-Medium | 0.0 | 67.3 | 73.8 | 0.0 | 0.0 | 0.0 | 40.9 | 73.8 | 0.0 | 4.0 | 2.0 | **92.4** | 6.7 ± 2.6 | **93.3± 2.6** |
| | Antmaze-Large | 0.0 | 20.5 | 30.3 | 0.0 | 0.0 | 0.0 | 37.5 | 30.3 | 0.0 | 0.0 | 27.3 | 68.0 | 26.7 ± 4.7 | **73.3± 4.7** |
| Average | | 0.0 | 43.6 | 53.0 | 0.0 | 0.0 | 0.0 | 48.1 | 53.0 | 0.0 | 3.0 | 13.3 | 77.6 | 21.7 | **83.3** |
| Kitchen | Mixed | 51.5 | 52.4 | 51.0 | 17.3 | 0.0 | 25.8 | 56.1 | 51.0 | 1.6 | 65.0 | 52.5 | **73.6** | 63.3 ± 1.1 | 72.7± 1.4 |
| | Partial | 38.0 | 50.0 | 46.3 | 6.7 | 35.5 | 31.4 | 37.4 | 63.3 | 1.6 | 57.0 | 55.7 | **74.4** | 65.0 ± 1.3 | 71.5± 1.7 |
| Average | | 44.8 | 51.2 | 48.7 | 12.0 | 17.8 | 28.6 | 46.8 | 57.2 | 1.6 | 61.0 | 54.1 | **74.0** | 64.2 | **72.1** |
| Maze2d | Large | 5.0 | 12.5 | 59.0 | -0.5 | 14.1 | 35.7 | 61.7 | 59.0 | **171.6** | 111.8 | 123.0 | 39.1 | 149.2 ± 7.5 | 44.3 ± 9.2 |
| | Medium | 30.3 | 5.0 | 32.8 | 19.1 | 68.5 | 31.7 | 34.1 | 50.2 | 111.7 | 103.7 | 121.5 | 32.2 | **128.2± 6.6** | 78.3 ± 10.4 |
| | U-Maze | 3.8 | 5.7 | 37.4 | -15.4 | 76.4 | 18.1 | 40.5 | 77.0 | 111.3 | 113.8 | 113.9 | 31.2 | **127.4± 1.2** | 78.2± 14.8 |
| Average | | 13.0 | 7.7 | 43.1 | 1.1 | 53.0 | 28.5 | 45.4 | 62.1 | **131.5** | 109.8 | 119.5 | 34.2 | **134.9** | 66.9 |

Transformer (DT) (Chen et al., 2021); data-augmented methods Synthetic experience replay (SER) (Lu et al., 2023) and DiffserStitch (Li et al., 2024); and diffusion-based planning methods including Diffuser (Janner et al., 2022), Decision stacks (Zhao & Grover, 2023), Decision Diffuser (DD) (Ajay et al., 2022), and DiffuserLite (Dong et al., 2024).

**Implementation Details.** For D-RAD, we follow the same planning horizon as Diffuser, while DL-RAD uses the horizon defined in DiffuserLite. More details about hyperparameter please refer Appendix C. Training was conducted on 4 NVIDIA A40 GPUs, an Intel Gold 5220 CPU, and 504GB memory, optimized with the Adam optimizer (Kingma & Ba, 2014). The baselines are implemented following their official implementations for a fair comparison.

To evaluate the effectiveness of the proposed RAD framework, we compare D-RAD and DL-RAD with representative baselines across different categories on D4RL. Results in Table 1 show that RAD achieves the best or near-best performance on 16 out of 18 datasets (RQ1). More specifically, (1) in the MuJoCo environments, on sub-optimal datasets (Medium and Medium-Replay), DL-RAD exhibits more pronounced improvements compared to existing methods. Notably, on the Walker-Medium-Replay dataset, RAD outperforms the highest-scoring baseline, DiffuserLite, by approximately 3. This improvement can be attributed to RAD's retrieval of *high-return and reachable* states from the offline dataset as target states. In sub-optimal datasets

with heterogeneous data quality, many low-return or sub-optimal trajectories exist, which may mislead conventional methods. By retrieving high-quality state segments, RAD effectively escapes from low-return regions, allowing the policy to learn more high-return behaviors during training, thereby improving performance. In contrast, on the Medium-Expert datasets, most trajectories are already near-optimal, and even without the retrieval mechanism, policies can learn high-return behaviors, resulting in limited marginal gains from retrieval; (2) in the AntMaze environments, RAD consistently outperforms all baseline methods across different datasets. For example, on AntMaze-Medium-Play, DL-RAD achieves a score of 86.7, surpassing the best-performing baseline, DiffuserLite (78.0), by approximately 8.7. On AntMaze-Large-Play, DL-RAD reaches 80.0, which is more than 7 higher than other methods. This indicates that RAD can effectively perform long-horizon planning under sparse reward conditions. By selecting high-return target states from expert trajectories and generating feasible action sequences, RAD guides the agent along a reasonable path toward the final goal; (3) in Maze2d, both D-RAD and DL-RAD surpass Diffuser and DiffuserLite, demonstrating that the retrieval-guided mechanism helps generate higher-quality long-horizon actions.

Additional evaluations on the OGBench stitching and explore settings are provided in Appendix D.10.

### 5.2. Generalization

Offline reinforcement learning faces the critical challenge of whether the learned policy can generalize to situations not

---

[2]Results for SER and DStitch are obtained by applying the methods with IQL as the offline RL algorithm.

*Table 2.* Performance under distribution shifts. Models are trained on Medium-Replay datasets and evaluated with initial states replaced by states from the corresponding Random datasets. The best results are in bold.

| Dataset | CQL | DT | MOPO | DiffStitch | DiffuserLite | DL-RAD |
|---|---|---|---|---|---|---|
| HalfCheetah | 37.9 | 26.3 | **62.3** | 26.4 | 37.6 | 39.2± 0.02 |
| Hopper | 63.6 | 35.7 | 39.5 | 37.6 | 60.4 | **90.2± 0.05** |
| Walker2d | 50.6 | 46.0 | 78.6 | 13.7 | 73.9 | **85.8± 0.03** |

*Table 3.* Performance under severe state distribution shifts. Left: visualization of the region-restricted setting, where the red boundary denotes the removed bottom-right region and the blue dots indicate evaluation initial states sampled from this unseen region. Right: quantitative results on AntMaze-Medium-Play and AntMaze-Large-Play. The coverage indicates the percentage of data from the bottom-right region retained in training. The best results are in bold.

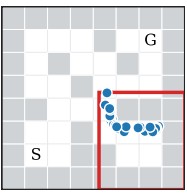

| Dataset | Coverage Setting | CQL | DT | MOPO | DiffuserLite | DL-RAD |
|---|---|---|---|---|---|---|
| AntMaze-Medium-Play | 0% coverage | 0.0 | 0.0 | 0.0 | 0.0 | **36.7 ± 5.1** |
| | 20% coverage | 0.0 | 0.0 | 0.0 | 0.0 | **52.2 ± 2.2** |
| AntMaze-Large-Play | 0% coverage | 0.0 | 0.0 | 0.0 | 0.0 | 0.0 ± 0.0 |
| | 20% coverage | 0.0 | 0.0 | 0.0 | 0.0 | **47.8 ± 1.1** |

present in the training dataset. To evaluate this, we first initialize the states by randomly sampling from the corresponding Random dataset. Subsequently, we leverage policies pre-trained on the Medium-Replay dataset to interact with the environment. As shown in Table 2, DL-RAD demonstrates clear improvements over DiffuserLite and other baselines in Hopper and Walker2d, while underperforming MOPO in HalfCheetah. We hypothesize that this performance gap arises because the HalfCheetah Medium-Replay dataset exhibits both higher average cumulative returns and richer trajectory diversity compared to Hopper and Walker2d (Shan et al., 2024), allowing MOPO to fully exploit them through dynamics modeling and thereby achieve superior performance. In this case, DL-RAD's retrieval mechanism provides limited additional benefits compared with dynamics modeling of MOPO. In Hopper and Walker2d, DL-RAD achieves substantially higher returns than all other baselines. This suggests that the retrieved target states enable the agent to better exploit trajectories in the offline dataset for decision making, thereby allowing the learned policy to generalize to new states not covered in the training dataset.

We further evaluate our method under more severe state distribution shifts using AntMaze-Medium-Play and AntMaze-Large-Play. For each dataset, we construct a region-restricted training set by removing all trajectories that enter the bottom-right region, and evaluate DL-RAD and the baselines from initial states sampled from that unseen region. We further consider a partial-coverage setting, where only 20% of the data in the bottom-right region is retained. For fairness, under each setting, both methods are trained on the same training set and evaluated from the same initial states. The results are shown in Table 3. These results suggest that DL-RAD is more robust than the baselines under severe state distribution shifts. On AntMaze-Medium-Play, DL-RAD achieves 36.7 under 0% coverage and 52.2 under

20% coverage, while all baselines fail in both settings. On AntMaze-Large-Play, all methods fail under 0% coverage, but under 20% coverage, DL-RAD reaches 47.8 whereas all baselines remain at 0.0. Overall, these results show that RAD can generalize to new states not covered in the training dataset(RQ2).

Additional out-of-distribution generalization evaluations are provided in Appendix D.5.

### 5.3. Ablation Study

To evaluate the contribution of each component in the RAD, we conduct ablation studies, with the results shown in Figure 3. Specifically, we have three variants:

- **-TS** removes the TS module, causing RAD to fall back to the backbone.

- **-TSR** removes the TS module and randomly samples targe states from the dataset.

- **-SE** removes the SE module and replaces the predicted transition horizon with a randomly selected number of steps.

- **-PTS** replaces the random offset $i \sim U(1, H-1)$ in the pseudo target strategy (4.4) with a fixed step $i$, so that the pseudo target is always selected at the same horizon within the trajectory.

Specifically, we observe the following : (1) Compared **-TSR**, **-TS** and **D-RAD**, **-TSR** performs worst, **-TS** is better, and **D-RAD** achieves the best performance in most cases. This trend confirms that both the TS and SE modules contribute positively (RQ3), as removing them (**-TS**) reduces performance, and further replacing TS with random targets (**-TSR**)

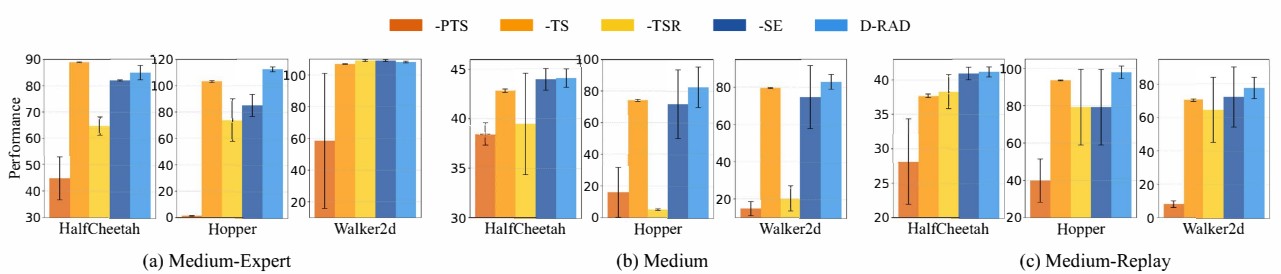

*Figure 3.* Results of ablation experiments on different variants.

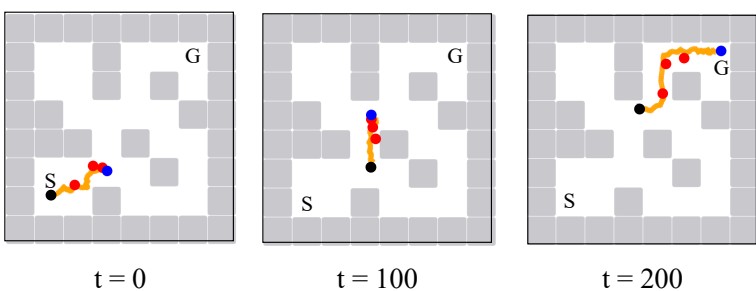

| t = 0 | t = 100 | t = 200 |

*Figure 4.* Visualization of policy predictions and real environment roll-outs in the AntMaze environment. Black dots denote the start states of the corresponding steps, blue dots indicate the target states ($S$ for the map's starting position and $G$ for the final goal), red dots represent the intermediate states, and the orange line represents the actual trajectory of agent-environment interaction over time, starting from the current moment.

degrades it even more. (2) The contrast between **-TSR** and **D-RAD** highlights that retrieving appropriate target states is crucial. While randomly injected targets introduce noise and mislead the policy, retrieved targets provide informative guidance that effectively directs the agent toward high-return regions. (3) **-PTS** performs worse than **D-RAD**. This validates the pseudo target strategy: a fixed horizon limits adaptability, whereas sampling random offsets across horizons enriches training and improves robustness.

The more additional ablation study results for **RAD** are provided in Appendix D.2 and Appendix D.3.

### 5.4. Visulization

To further investigate the target states generated by RAD, we conducted a visualization experiment. Specifically, we selected the AntMaze-Medium-Replay and visualized part of the target states generated by the DL-RAD agent, along with the actual trajectories obtained through environment interactions.

The results are shown in Fig. 4. From the figure, we can observe the following (RQ4): (1) The target states generated by the policy are located at reasonable positions, indicating that the target states are reasonable. (2) Guided by these target states, the agent can successfully reach the final goal $G$, demonstrating that enhancing the decision-making with the guidance of target states is effective. (3) The actual trajecto-

ries obtained from environment interactions align well with the generated target states, suggesting that the targets are not only theoretically reasonable but also practically achievable, thereby validating the reachability of our method.

### 5.5. Computational Cost

A potential concern with RAD is the additional computational cost introduced by its auxiliary modules. During training, the extra cost mainly comes from learning the SE module. During inference, the additional overhead comes from the retrieval component in the TS module, which performs vector-based similarity search.

For training, RAD has the same backbone training time as the diffusion-policy baseline. The only additional cost comes from training the SE module. However, this module can be trained in parallel with backbone training, therefore the additional practical overhead is limited. We report the detailed training time of DL-RAD in Table 4.

*Table 4.* Training time of DL-RAD.

| Dataset | Backbone Training | SE-module Training |
|---|---|---|
| Navigation | 25h | 4h |
| Locomotion | 8.4h | 3h |

To provide an accurate measurement of the actual overhead

in RAD, we have measured the total time taken by RAD to generate a single action in inference, including the time for the policy to compute the action from a given state and the time required to execute the action in the environment and transition to the next state. The results are summarized in Table 5.

*Table 5.* Per-step inference time for different methods.

| Environment | Method | Inference Time (s) | Performance |
|---|---|---|---|
| AntMaze | DiffuserLite | 0.06 | 77.6 |
| AntMaze | DL-RAD | 0.26 | 83.3 |
| Locomotion | DiffuserLite | 0.05 | 86.0 |
| Locomotion | DL-RAD | 0.11 | 87.5 |

Compared to DiffuserLite, DL-RAD introduces an additional overhead (approximately +0.20s in AntMaze and +0.06s in Locomotion), mainly due to the retrieval component in the TS module. The observed latency is acceptable for real-time execution in these tasks, given the performance improvement DL-RAD achieves (RQ5). Moreover, since consecutive states are often highly similar (Tang et al., 2024), RAD can further reduce the accumulated inference cost by performing target selection periodically rather than at every step. Additional experiments on periodic replanning and fixed-horizon wall-clock time are provided in Appendix D.8.

## 6. Conclusion and Discussion

We presented RAD, a retrieval-augmented method for offline RL. RAD improves generalization by dynamically retrieving high-return states as target states and leveraging diffusion-based trajectory generation for planning. By conditioning on these target states, the agent is guided toward reachable high-return regions, gradually escaping low-return and poorly covered areas, and thereby generalizing to previously unseen states. Experiments show that RAD matches or outperforms prior methods across diverse settings. This demonstrates the potential of our retrieval-augmented mechanism in overcoming data coverage limitations in offline RL. However, RAD still relies on the coverage of the offline dataset. When high-return target states are absent, the planning based on the target states is ineffective, thereby affecting planning performance.

Our current experiments are primarily conducted on standard D4RL tasks, with visualizations and analyses limited to a few environments. Future work could extend RAD to more complex and diverse scenarios. In addition, we plan to investigate more efficient retrieval strategies to further improve the applicability and effectiveness of RAD.

## Acknowledgements

This work was supported by the National Natural Science Foundation of China (Grants No.62307020, No,U2341229), the Fundamental and Interdisciplinary Disciplines Breakthrough Plan of the Ministry of Education of China (Grant No.JYB2025XDXM903), and the New Cornerstone Science Foundation through the XPLORER PRIZE.

## Impact Statement

This paper presents work whose goal is to advance the field of Machine Learning. There are many potential societal consequences of our work, none which we feel must be specifically highlighted here.

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

# A. Details of Experimental Setup

Gym-MuJoCo (Brockman et al., 2016) on D4RL consists of three popular offline RL locomotion tasks (Hopper, HalfCheetah, Walker2d). These tasks require controlling three MuJoCo robots to achieve maximum movement speed while minimizing energy consumption under stable conditions. D4RL provides three different quality levels of offline datasets: "medium" containing demonstrations of medium-level performance, "medium-replay" containing all recordings in the replay buffer observed during training until the policy reaches medium performance, and "medium-expert" which combines medium and expert level performance equally. We further analyze the returns distribution of these datasets, showing the differences in trajectory quality among the Medium, Med-Replay, and Med-Expert datasets for HalfCheetah, Hopper, and Walker2d (Figure 5).

FrankaKitchen (Gupta et al., 2019) requires controlling a realistic 9-DoF Franka robot in a kitchen environment to complete several common household tasks. In offline RL testing, algorithms are often evaluated on "partial" and "mixed" datasets. The former contains demonstrations that partially solve all tasks and some that do not, while the latter contains no trajectories that completely solve the tasks. Therefore, these datasets place higher demands on the policy's "stitching" ability. During testing, the robot's task pool includes four sub-tasks, and the evaluation score is based on the percentage of tasks completed.

AntMaze (Fu et al., 2020) requires controlling the 8-DoF "Ant" quadruped robot in MuJoCo to complete maze navigation tasks. In the offline dataset, the robot only receives a reward upon reaching the endpoint, and the dataset contains many trajectory segments that do not lead to the endpoint, making it a difficult decision task with sparse rewards and a long horizon. The success rate of reaching the endpoint is used as the evaluation score, and common model-free offline RL algorithms often struggle to achieve good performance.

Maze2D (Fu et al., 2020) is a navigation task in which a 2D agent needs to traverse from a randomly designated start location to a fixed goal location where a reward of 1 is given. No reward shaping is provided at any other location. The objective of this task is to evaluate the ability of offline RL algorithms to combine previously collected sub-trajectories in order to find the shortest path to the evaluation goal. Three maze layouts are available: "umaze", "medium", and "large". The expert data for this task is generated by selecting random goal locations and using a planner to generate sequences of waypoints that are followed by using a PD controller to perform dynamic tracking.

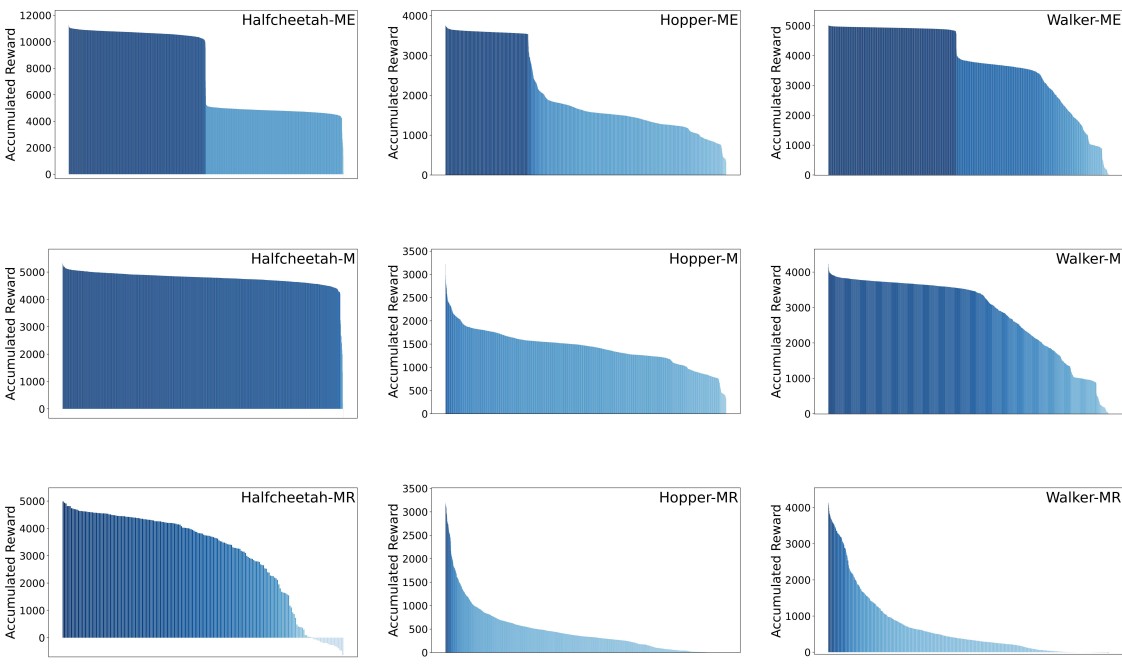

*Figure 5.* Returns distribution of Med-Expert, Medium and Med-Replay datasets of Halfcheetah, Hopper, Walker2d.

# B. Algorithms.

---

**Algorithm 1** Training

---

**Require:** Offline dataset $\mathcal{D}$, batch size $B$, horizon $H$, diffusion steps $K$, diffusion model $\phi_\theta$, guidance model $\mathcal{J}_\phi$, step estimation model $f_e$,

 1: **for** each training iteration **do**
 2:     Sample a batch of trajectories $\{\tau_t\}_{t=1}^B$ from $\mathcal{D}$
 3:     **for** each trajectory $\tau_t$ in batch **do**
 4:         Randomly select an offset $i \sim \mathcal{U}(1, H-1)$
 5:         Set $\boldsymbol{s}_{t+i}$ as pseudo target
 6:         Apply forward diffusion on sub-trajectory $\tau_t$ to obtain noisy $\hat{\tau}_t^k$
 7:         Compute generation loss: $\mathcal{L}_d$ by Eq. 12.
 8:         Compute guidance loss: $\mathcal{L}_g$ by Eq. 13.
 9:         Compute step estimation loss: $\mathcal{L}_e$ by Eq. 14.
10:     **end for**
11:     Optimize $\phi_\theta$, $\mathcal{J}_\phi$, and $f_e$ independently with $\mathcal{L}_d$, $\mathcal{L}_g$, and $\mathcal{L}_e$, respectively.
12: **end for**

---

**Algorithm 2** Planning and decision-making

---

**Require:** Current state $\boldsymbol{s}_t$, diffusion model $\phi_\theta$, step estimation model $f_e$, database $\mathcal{D}$, similarity threshold $\delta$, Top-$k$ selection $k$

 1: Retrieve candidate states $\{\boldsymbol{s}_i'\}$ from $\mathcal{D}$ s.t. $\text{sim}(\boldsymbol{s}_t, \boldsymbol{s}_i') \geq \delta$
 2: **if** candidates found **then**
 3:     Select top-$k$ states by similarity and form $\mathcal{C}_s$
 4:     Extract high-return candidate states $\mathcal{C}_g$ by Eq. 3
 5:     Select target state $\boldsymbol{s}_t^g$ from $\mathcal{C}_g$ by Eq. 4
 6:     Estimate step $\hat{i}_t$ by Eq. 6
 7: **else**
         fall back to the backbone planner
 8: **end if**
 9: Initialize noisy sub-trajectory $\tau_{temp} \sim \mathcal{N}(0, I)$ of length $H \geq \hat{i}_t$
10: Substitute $\boldsymbol{s}_t$ and $\boldsymbol{s}_t^g$ into $\tau_{temp}$ at positions 0 and $\hat{i}_t$
11: **for** $k = K$ down to 1 **do**
12:     Reverse denoise $\tau_{temp}$ using $\phi_\theta$ and guidance $\mathcal{J}_\phi$
13: **end for**
14: Obtain clean trajectory $\hat{\tau}_t^0$
15: **if** trajectory contains state-action pairs **then**
16:     Execute action $\hat{\boldsymbol{a}}_t$ in environment
17: **else**
18:     Use inverse dynamics: $\boldsymbol{a}_t = f_a(\boldsymbol{s}_t, \hat{\boldsymbol{s}}_{t+1})$ and execute
19: **end if**

---

# C. Implementation Details

- We represent the noise model in D-RAD with a temporal U-Net (Janner et al., 2022), consisting of a U-Net structure with 6 repeated residual blocks. Each block consisted of two temporal convolutions, each followed by group norm , and a final Mish nonlinearity. Timestep and condition embeddings, both 128-dimensional vectors, are produced by separate 2-layered MLP (with 256 hidden units and Mish nonlinearity) and are concatenated together before getting added to the activations of the first temporal convolution within each block .

- In DL-RAD, we utilize DiT (Peebles & Xie, 2023) as the neural network backbone for all diffusion models and rectified flows, with an embedding dimension of 256, 8 attention heads, and 2 DiT blocks.

*Table 6.* Planning horizons and levels used in D-RAD and DL-RAD across different environments.

| Method | Environment | Planning Horizon $H$ | Temporal Jumps / Levels |
|---|---|---|---|
| D-RAD | MuJoCo (locomotion) | 32 | - |
| D-RAD | Kitchen | 32 | - |
| D-RAD | AntMaze | 64 | - |
| D-RAD | Maze2D U-Maze | 128 | - |
| D-RAD | Maze2D Medium | 265 | - |
| D-RAD | Maze2D Large | 384 | - |
| DL-RAD | Kitchen | 49 | 16, 4, 1 |
| DL-RAD | MuJoCo (locomotion) | 129 | 32, 8, 1 |
| DL-RAD | AntMaze | 129 | 32, 8, 1 |

- For all locomotion tasks, we regard trajectories of length 1000 as expert demonstrations. For AntMaze, expert targets are selected from trajectories whose first hitting step of the goal lies between 150 and 600. For Kitchen, trajectories that successfully complete three designated tasks are considered expert. For Maze2D, expert trajectories are selected according to the number of steps required to reach the goal for the first time: 400-600 steps for Maze2D-medium, 400-800 steps for Maze2D-large, and 200-300 steps for Maze2D-umaze.

- The planning horizons and temporal jumps used in D-RAD and DL-RAD across different environments are summarized in Table 6.

- We consider the top-6 most similar candidates when selecting the target state in D-RAD, and the top-500 most similar candidates when selecting the target state in DL-RAD.

## D. Additional Experiment Results

### D.1. Efficiency

A potential concern with RAD is the additional memory footprint introduced by the retrieval component in the TS module, which stores vector-based indices for similarity search and may also introduce retrieval overhead during inference.

To characterize the worst-case memory footprint, we measured the cost of storing high-dimensional indices representing the entire state space. This full-state index requires approximately 1.4 GB of CPU memory on our in-house server (502 GB RAM), which remains negligible at system scale. For the same full-state index, a single retrieval including network latency takes about 0.4 seconds, and this cost can be further reduced through local caching.

However, RAD does not query the entire state space during evaluation. All retrieval operations in our actual experiments are performed within a pre-constructed expert database, which is substantially smaller. Consequently, the true time overhead differs from the 0.4-second worst-case measurement above.

### D.2. Effect of Target-State Guidance

We also conduct ablation studies on DL-RAD to further examine the contribution of retrieval-guided target conditioning. First, we remove the target-state guidance from DL-RAD and compare the results. Table 7 summarizes the results across HalfCheetah-medium-replay, Hopper-medium-replay and Walker2d-medium-replay. The version without target-state guidance shows a clear drop in performance compared with the full DL-RAD model.

*Table 7.* Ablation study comparing DL-RAD with and without target states across different environments.

| Environment | No Target States | DL-RAD |
|---|---|---|
| HalfCheetah-MR | 41.9 | 44.4 |
| Hopper-MR | 24.1 | 100.4 |
| Walker2d-MR | 72.4 | 93.5 |

## D.3. Comparison with DiffuserLite and Direct Retrieval Combination

To further investigate the role of the proposed retrieval-guided design, we compare DL-RAD with two related variants: DiffuserLite and a direct retrieval combination of DiffuserLite (DRC), and evaluate all methods under the same region-restricted AntMaze setting used in the generalization evaluation (section 5.2), where trajectories entering the bottom-right region are removed from the training set and evaluation initial states are sampled from that region. DiffuserLite generates trajectories only from the current state and the return condition, without using any retrieved target state. DRC augments DiffuserLite with a naive retrieval mechanism: given the current state, it first retrieves a state whose two-dimensional position is close to the current state, and then inject the retrieved state as the target condition for planning. However, DRC does not use the TS and SE models of RAD, and therefore the retrieved state may not provide reliable or well-aligned guidance for planning. In contrast, DL-RAD retrieves high-return and reachable target states and conditions the planner on these targets, so that the generated trajectory is explicitly guided toward a more promising region. As shown in Table 8, the result indicates that the performance gain of DL-RAD does not simply come from adding retrieval to DiffuserLite.

*Table 8.* Comparison among DiffuserLite, DRC, and DL-RAD, under the region-restricted setting. The 0% and 20% settings denote the amount of data from the bottom-right region retained during training.

| Dataset | Coverage Setting | DiffuserLite | DRC | DL-RAD |
|---|---|---|---|---|
| AntMaze-M-P | 0% coverage | 0.0±0.0 | 13.3 ± 3.8 | **36.7 ± 5.1** |
| AntMaze-M-P | 20% coverage | 0.0±0.0 | 15.6 ± 8.0 | **52.2 ± 2.2** |
| AntMaze-L-P | 0% coverage | 0.0±0.0 | 0.0 ± 0.0 | 0.0 ± 0.0 |
| AntMaze-L-P | 20% coverage | 0.0±0.0 | 20.0 ± 10.0 | **47.8 ± 1.1** |

## D.4. Parameter Study

To investigate the effect of the minimum similarity threshold $\delta$ in the target selection module, we conduct experiments varying $\delta$ while keeping other settings fixed. The results are summarized in Table 9 and Table 10.

*Table 9.* Effect of minimum similarity threshold $\delta$ for D-RAD.

| $\delta$ | HalfCheetah-M | Hopper-M | Walker2d-M |
|---|---|---|---|
| 0.0 | 43.6 | 54.2 | 64.3 |
| 0.5 | 43.7 | 77.5 | 53.2 |
| 0.8 | 44.0 | 74.8 | 58.4 |
| 0.9 | 44.2 | 82.5 | 82.8 |

*Table 10.* Effect of minimum similarity threshold $\delta$ for DL-RAD.

| $\delta$ | AntMaze-L-P | Kitchen-M | Maze2d-M | Hopper-MR |
|---|---|---|---|---|
| 0.6 | 60.7 | 0.0 | 52.0 | 100.4 |
| 0.7 | 50.0 | 2.5 | 59.0 | 100.3 |
| 0.8 | 80.0 | 60 | 78.3 | 100.2 |
| 0.9 | 70.0 | 72.7 | 60.1 | 96.5 |

## D.5. Additional distribution shifts Experiments

To more systematically evaluate whether the learned policies can generalize to states not present in the training dataset, we conducted additional OOD tests on both Maze2D and AntMaze. For Maze2D, we randomly sampled initial states from maze2d-open-v0 and executed policies trained on maze2d-umaze-v1 or maze2d-medium-v1. For AntMaze, we randomly sampled initial states from AntMaze-Medium-Diverse-v2 and evaluated policies trained on AntMaze-Medium-Play-v2 or AntMaze-Large-Play-v2. The results are reported in Table 11.

Across all datasets, DL-RAD consistently outperforms the baselines, often by a substantial margin. This demonstrates the effectiveness of our method.

*Table 11.* Performance under distribution shifts.

| Environment | DiffStitch | DiffuserLite | DL-RAD |
|---|---|---|---|
| Antmaze-Medium-Play | 36.7 | 6.7 | 43.3 |
| Antmaze-Large-Play | 33.3 | 0.0 | 36.7 |
| Maze2D Medium | 14.7 | 28.3 | 38.0 |
| Maze2D U-maze | 10.8 | 28.5 | 44.7 |

*Table 12.* Sensitivity to Retrieval Ranking Quality.

| Environment | Top-1 | Top-3 | Top-5 | Top-7 |
|---|---|---|---|---|
| Antmaze-Medium-Play | 86.7 | 85.3 | 85.3 | 72.0 |
| Antmaze-Large-Play | 80.0 | 73.3 | 70.0 | 66.7 |
| Antmaze-Medium-Diverse | 93.3 | 86.7 | 84.3 | 83.3 |
| Antmaze-Large-Diverse | 73.3 | 62.0 | 58.7 | 62.0 |

### D.6. Sensitivity to Imperfect Ranking in the Retrieval Module

To examine whether RAD depends heavily on perfect ranking within the retrieval module, we conducted an additional stress test on AntMaze by deliberately degrading the ranking quality. In the final step of the TS module, instead of always selecting the top-1 state, we constructed candidate sets of size 1, 3, 5, and 7, corresponding to increasingly noisy retrieval. For each candidate set, we randomly sampled one state as the retrieved target, thereby simulating scenarios in which the retrieval mechanism returns suboptimal or partially misranked states. The results are summarized in Table 12.

As the candidate set grows larger, the noise in the retrieval ranking increases, and the performance shows a gradual downward trend. This behavior is expected: when the retrieved target state is not necessarily the optimal one, the guidance provided to the planner becomes weaker, leading to reduced success rates. More importantly, however, this degradation is gradual rather than catastrophic. Comparing these results against the baselines in Table 1, we observe that even the worst Top-7 performance remains competitive in most environments. For example, in AntMaze-Medium-Diverse, the Top-7 setting still achieves 83.3, ranking among the top three methods.

Therefore, although imperfect ranking introduces some negative effects on the performance, RAD can still benefit from the retrieved target even when it is suboptimal, as long as the retrieved state lies within a reasonably high-value region.

### D.7. Sensitivity to Similarity Metric Choice

We note that a different evaluation metric was adopted in our previous implementation. To ensure consistency and reproducibility, we follow the same metric in this paper. As shown in Tables 13, the performance gap between the two metrics is small across all evaluated tasks. The results show that RAD is compatible with multiple metric choices.

*Table 13.* Sensitivity to similarity metrics on navigation and locomotion tasks.

| Domain | Task | Cosine | Euclidean |
|---|---|---|---|
| Navigation | Antmaze-Medium-Play | 83.3 | 86.7 |
| Navigation | Antmaze-Large-Play | 76.7 | 80.0 |
| Locomotion | HalfCheetah-medium-replay | 44.4 | 44.0 |
| Locomotion | HalfCheetah-medium | 48.8 | 48.3 |
| Locomotion | HalfCheetah-medium-expert | 90.1 | 88.6 |

### D.8. Computational Cost and Inference Efficiency

For inference, in our original evaluation, target selection was performed after every interaction with the environment, i.e., once a new state was observed, a new target state was retrieved for planning. However, since consecutive states are often highly similar (Tang et al., 2024), target selection does not necessarily need to be performed at every step. Instead, it can be carried out once every several steps, and the same retrieved target can be reused for planning over the intermediate similar states within that interval. Therefore, this issue can be mitigated by performing planning periodically rather than at every

step, thereby reducing the accumulated inference cost. However, periodic planning may also lead to some performance degradation. To verify whether RAD can preserve performance while reducing inference cost, we conducted additional experiments. Specifically, we compared the full-rollout wall-clock time and performance of RAD under two settings: replanning every 10 environment steps and replanning at every step. The results are summarized in Table 14.

*Table 14.* Full-rollout wall-clock time and performance under different replanning intervals.

| Environment | Replanning Interval | Score | Full-rollout Inference Time (s) | Time Reduction vs. 1-step Replanning |
|---|---|---|---|---|
| AntMaze-Medium-Play | 1 step | 86.7 | $147.1 \pm 48.5$ | – |
| AntMaze-Medium-Play | 10 steps | 86.7 | $33.6 \pm 20.1$ | 113.5 (77.2%) |
| Hopper-medium-replay | 1 step | 100.4 | $109.9 \pm 1.1$ | – |
| Hopper-medium-replay | 10 steps | 99.8 | $43.2 \pm 0.8$ | 66.7 (60.7%) |

These results show that periodic replanning substantially reduces the full-rollout inference time while causing only a negligible performance change. On AntMaze-Medium-Play, increasing the replanning interval from 1 step to 10 steps reduces the full-rollout inference time from $147.1 \pm 48.5$s to $33.6 \pm 20.1$s, i.e., a 77.2% reduction, while the score stays the same at 86.7. On Hopper-medium-replay, the full-rollout inference time drops from $109.9 \pm 1.1$s to $43.2 \pm 0.8$s, i.e., a 60.7% reduction, while the score only decreases slightly from 100.4 to 99.8. For the large variance on AntMaze, it mainly comes from the benchmark protocol rather than RAD itself: successful episodes often terminate early after reaching the goal, while failures may run the full 1000 steps, naturally leading to high wall-clock variance. This is also seen in the baseline, where DiffuserLite on AntMaze-Medium-Play shows $19.1 \pm 11.2$s rollout time. Overall, these results provide direct evidence that periodic replanning can substantially mitigate RAD's additional inference latency in practice, with almost no performance degradation.

To disentangle inference cost from rollout length, we additionally measured the wall-clock time over the same environment steps under different replanning intervals. The results are shown in Table 15. Under a fixed rollout horizon, periodic replanning substantially reduces RAD's inference overhead, bringing its wall-clock time close to DiffuserLite.

*Table 15.* Wall-clock time over the same environment steps under different replanning intervals.

| Environment | RAD (interval = 1) | RAD (interval = 10) | DL |
|---|---|---|---|
| AntMaze-Medium-Play | $37.2 \pm 5.5$s | $16.0 \pm 0.8$s | $13.1 \pm 0.5$s |
| AntMaze-Large-Play | $23.0 \pm 0.4$s | $9.4 \pm 0.2$s | $8.5 \pm 0.3$s |

### D.9. Comparison with Highest-Return-Only Target Selection

We clarify why the target is selected based on the longest remaining trajectory rather than alternatives such as the highest-return trajectory. RAD does not rely on trajectory length alone. We first filter for high-return states, then select the one with the longest remaining trajectory among them. This favors earlier entry into high-value segments rather than isolated late states, providing a more stable anchor for diffusion planning and enough future trajectory support for long-horizon tasks. The comparison with a highest-return-only strategy is shown in Table 16, which further validates this design.

*Table 16.* Comparison between highest-return-only selection and high-return-filtered longest-remaining-trajectory selection.

| Environment | Highest-return-only | High-return-filtered longest-remaining-trajectory selection(DL-RAD) |
|---|---|---|
| HalfCheetah-Medium-Replay | 43.9 | 44.4 |
| HalfCheetah-Medium | 48.2 | 48.8 |
| HalfCheetah-Medium-Expert | 89.9 | 90.1 |
| AntMaze-Medium-Play | 73.3 | 86.7 |
| AntMaze-Medium-Diverse | 77.8 | 93.3 |
| AntMaze-Large-Play | 51.1 | 80.0 |
| AntMaze-Large-Diverse | 60.0 | 73.3 |

## D.10. Additional Evaluation on OGBench

In this work, we focus on the problem of how an agent can make better decisions to achieve long-term high returns when it encounters out-of-distribution (OOD) states. This objective is slightly different from the stitching capability evaluated in OGBench, which primarily emphasizes explicitly composing trajectories from disconnected sub-trajectories. Therefore, in our main experiments, we conduct experiments under a modified D4RL setting, where we train the policy on the standard D4RL datasets (expert, medium, and medium-replay), and evaluate it by initializing from states randomly sampled from the D4RL random dataset. This setup introduces a clear distribution shift, requiring the agent to generalize beyond the training distribution and make coherent decisions from unfamiliar states.

We also evaluate RAD on the stitching and explore settings of OGBench (Park et al., 2025). The results are shown in Table 17. RAD performs best in both settings, showing that RAD is effective not only in the more challenging OOD settings emphasized in our paper, but also on standard benchmarks specifically designed for trajectory stitching.

Table 17. Evaluation on the stitching and explore settings of OGBench.

| Setting | Dataset | GCBC | GCIVL | GCIQL | QRL | CRL | HIQL | RAD |
|---|---|---|---|---|---|---|---|---|
| Stitch | AntMaze-Medium | 45 | 44 | 29 | 59 | 53 | 94 | **97** |
| | AntMaze-Large | 3 | 18 | 7 | 18 | 11 | 67 | **77** |
| Explore | AntMaze-Medium | 2 | 19 | 13 | 1 | 3 | 37 | **97** |
| | AntMaze-Large | 0 | 10 | 0 | 0 | 0 | 4 | **30** |

## D.11. Effect of Expert-Subset Filtering

To examine whether RAD's gains are mainly due to expert-subset filtering rather than retrieval, we train BC, IQL, and DiffuserLite on the exact same expert subset used to build RAD's retrieval database. RAD itself remains unchanged. The results are shown in Table 18. These results show that the gains cannot be explained by expert-subset filtering alone.

Table 18. Comparison with baselines trained on the same expert subset used to build RAD's retrieval database.

| Environment | BC (Full) | BC (Exp) | IQL (Full) | IQL (Exp) | DL (Full) | DL (Exp) | RAD |
|---|---|---|---|---|---|---|---|
| AntMaze-Medium-Play | 0.0 | 50.0 | 65.8 | 46.0 | 78.0 | 46.7 | **86.7** |
| AntMaze-Large-Play | 0.0 | 13.3 | 42.0 | 30.0 | 72.0 | 40.0 | **80.0** |

# E. Proof of Entropy Reduction With Target Conditioning

To analyze the effect of conditioning on additional target information in trajectory forecasting, we denote the subsequent trajectory generated for planning as a random variable $\tau$, the current state as $s_t$, and the retrieved target state as $s_g$. The predictive uncertainty associated with the trajectory given only $s_t$ is quantified by the conditional entropy $H(\tau \mid s_t)$; larger values indicate greater uncertainty and lower predictive confidence.

When the target state $s_g$ is included as an additional conditioning variable, the uncertainty becomes $H(\tau \mid s_t, s_g)$. By applying the chain rule of conditional entropy to the joint variable pair $(\tau, s_g)$, we obtain two equivalent expressions:

$$H(\tau, s_g \mid s_t) = H(\tau \mid s_t) + H(s_g \mid \tau, s_t) = H(s_g \mid s_t) + H(\tau \mid s_t, s_g).$$

Equating the two decompositions and reorganizing terms gives:

$$H(\tau \mid s_t) - H(\tau \mid s_t, s_g) = H(s_g \mid s_t) - H(s_g \mid \tau, s_t).$$

The right-hand side corresponds to the conditional mutual information $I(\tau; s_g \mid s_t)$, leading to:

$$H(\tau \mid s_t) - H(\tau \mid s_t, s_g) = I(\tau; s_g \mid s_t).$$

Since conditional mutual information is non-negative, i.e.,

$$I(\tau; s_g \mid s_t) \geq 0,$$

we obtain the inequality:

$$H(\tau \mid s_t) \geq H(\tau \mid s_t, s_g).$$

This result demonstrates that conditioning on the retrieved target state preserves or decreases the entropy of the trajectory distribution. Therefore, whenever $I(\tau; s_g \mid s_t) > 0$, the introduction of $s_g$ provides additional relevant information that reduces uncertainty and leads to more accurate and reliable trajectory prediction (i.e.the generation of the subsequent trajectory for planning).

## F. Comparison With Trajectory Stitching Methods

For completeness, we provide a detailed discussion on how RAD relates to trajectory-stitching approaches, particularly DiffStitch. RAD is indeed conceptually related to DiffStitch, as both methods are built based on generative models and conduct stitching. However, RAD is different DiffStitch due to:

- DiffStitch is a data augmentation method. It generates a fixed, enlarged data by stitching trajectory segments in the original offline dataset to enhance the offline dataset.

- RAD is an offline RL algorithm. It dynamically retrieves reachable and high-return states as the target states, and uses a diffusion model to plan toward the target states. This enables adaptive high-return-aware planning and decision making.

- Theoretically, the augmented data produced by DiffStitch can be further used to train RAD. This means the two approaches are compatible and can be organically combined to yield more efficient decision-making, rather than being mutually exclusive.

A structured comparison is provided below:

*Table 19.* Comparison between DiffStitch and RAD.

| Aspect | DiffStitch | RAD |
|---|---|---|
| Type | Diffusion-based data augmentation for offline RL | Diffusion-based offline RL method |
| Stitching | Yes, stitching trajectory segments for data generation | Yes, stitching the current state to the target state for planning |
| Trajectory Planning | N/A | Yes |
| Handling OOD States | Limited(fixed dataset) | Flexible via dynamic retrieval |
| Adaptivity | Static | Dynamic, per-step planning |
| Target | high-return subtrajectory | high-return states |

