# OpenReview forum: "RAD: Retrieval High-quality Demonstrations to Enhance Decision-making"
_ICML.cc/2026/Conference — ICML 2026 regular_

### Official Review · Reviewer_aXSD · 2026-02-28

**Soundness:** 2
**Presentation:** 2
**Significance:** 2
**Originality:** 2
**Overall Recommendation:** 3
**Confidence:** 4

**Summary:**

This paper proposes RAD, a retrieval-augmented framework for offline reinforcement learning.  The authors point out that traditional offline RL is limited by static data distributions, and Prior solutions based on synthetic data augmentation often fail to generalize to unseen scenarios in the (augmented) dataset. To address this, RAD retrieves high-return and reachable states from the offline dataset as target states, and leverages a generative model to generate sub- trajectories conditioned on these targets for planning.

**Compliance With Llm Reviewing Policy:**

Affirmed.

**Final Justification:**

After reading the rebuttal, I increased my score from 2 to 3. The additional experiments and clarifications, especially the expert-subset baseline and the periodic replanning results, partially addressed my earlier concerns and improved my confidence in the technical soundness of the work.

That said, I still do not find the contribution strong enough for ICML acceptance. In particular, RAD still feels more like a plug-and-play retrieval-based enhancement built on top of existing diffusion planning frameworks than a fully standalone offline RL method, which limits the strength of the novelty claim. In addition, the practicality argument is still not entirely convincing, since the reported full-rollout wall-clock time appears to conflate inference cost with rollout length and exhibits very large variance on AntMaze. Overall, I view the paper as technically reasonable and improved after rebuttal, but the contribution is still somewhat incremental relative to the bar for acceptance.

**Key Questions For Authors:**

1. The target selection module relies on similarity measures specific to the domain. Could the authors clarify how sensitive RAD is to the choice of similarity measures? In particular, if the similarity measures are not perfect, would RAD still be effective?

2. Why did the authors choose Diffuser and DiffuserLite as the generative backbones for RAD? The reasons for this choice should be verified through further discussion or experiments.

3.  RAD seems to be also a method of data augmentation. What are the differences and connections between it and DiffStitch?

4. The author has provided the total time taken by RAD to generate a single action. However, in long-horizon tasks, this cost accumulates over many steps, potentially leading to substantial latency in practice. In the case of a long-term situation, is the resulting delay acceptable?

5. The experimental evaluation is primarily conducted on standard D4RL-style benchmarks (Gym-MuJoCo, AntMaze, Maze2D, FrankaKitchen), which largely consist of fixed and well-curated offline datasets. While these benchmarks are widely adopted in offline RL, they may not sufficiently stress-test the claimed generalization capability of RAD under significant distribution shift.

**Limitations:**

Yes

**Strengths And Weaknesses:**

### Strengths:
1. The paper clearly articulates the limitations of static dataset augmentation in offline RL and motivates a retrieval-based augmentation that adapts at decision time.

2. Integrating a retrieval mechanism with a diffusion-based planning model for offline RL represents a advance over static or purely generative augmentation schemes.

### Weaknesses:
1. There are some aspects that affect the readability of the paper, such as what is meant by reachability and how is reachability quantified?

2. This paper mentions that retrieval is performed in a pre-constructed expert database, which is significantly smaller than the entire state space. However, the process of constructing this expert database is not explicitly described.

3. The novelty of the proposed method appears limited. Prior work has explored model-based trajectory stitching for offline data augmentation, and diffusion models have already been used for dataset enhancement. Additionally, generalization beyond offline datasets has been studied independently. The current work seems to mainly integrates these existing directions. This is a good paper, but it is not suitable for publication in ICML.

---

> ### Author Rebuttal · Authors · 2026-03-31
>
> Q1. What is meant by reachability in RAD, and how is it quantified?
>
> R1. In RAD, reachability refers to whether a retrieved target state can be feasibly reached from the current state within a reasonable number of steps. For example, a target state is reachable if the planner can generate a plausible sub-trajectory from the current state to that target. As for quantification, reachability is not explicitly formalized in our current framework; instead, we provide qualitative evidence through the visualization results in Section 5.5.
>
> Q2. The process of constructing this expert database is not explicitly described.
>
> R2. We describe the construction of the expert database in the first paragraph of Section 4.1, and provide the implementation details for the database in each dataset in the third paragraph of Appendix C.
>
> Q3. The novelty appears limited, as the method mainly combines existing ideas from data augmentation, diffusion-based enhancement, and offline RL generalization.
>
> R3. We would like to clarify several points regarding the novelty of our work：
> - Generalization beyond offline datasets is a research problem, not a novelty claim per se. Many prior studies have highlighted its importance, yet existing methods still struggle to generalize effectively to unseen regions. This challenge is precisely why we continue to study it: not because we claim novelty in identifying the problem, but because we aim to address its unresolved aspects.
> - While RAD does leverage diffusion models to address generalization beyond offline datasets, its novelty does not lie in diffusion models per se. The core contribution is the introducing of a retrieval mechanism to guide the diffusion-based policy during planning, enabling effective generalization to unseen regions. Introducing retrieval into offline RL planning has not been explored in prior work.
> - Regarding the comment on data augmentation: RAD is not designed as a data augmentation method.
>
> Q4. How sensitive is RAD to the choice and quality of the similarity metric?
>
> R4. Please refer to our response to Reviewer HuaD, R2
>
> Q5. Why were Diffuser and DiffuserLite chosen as the generative backbones for RAD?
>
> R5. We chose Diffuser and DiffuserLite because they are two strong yet substantially different diffusion-based planning backbones. Diffuser generates state-action trajectories,whereas DiffuserLite generates state-only trajectories and recovers actions via an inverse dynamics model. Using both allows us to verify that RAD’s retrieval-augmented mechanism is effective across different planning formulations, rather than being tied to a single generator design. Since the novelty of RAD does not lie in the generative model itself, adopting standard backbones helps isolate the performance gains brought by the retrieval component.
>
> Q6. What are the differences and connections between RAD and DiffStitch?
>
> R6. As discussed in Appendix F, RAD differs from DiffStitch due to:
> - DiffStitch is a data augmentation method. It generates a fixed data by stitching trajectory segments in the original offline dataset to enhance the offline dataset.
> - RAD is an offline RL algorithm. It dynamically retrieves reachable and high-return states as the target states, and uses a diffusion model to plan toward the target states.This enables adaptive high-return-aware planning and decision making.
> - Theoretically, the augmented data produced by DiffStitch can be further used to train RAD. This means the two approaches are compatible and can be organically combined to yield more efficient decision-making,rather than being mutually exclusive.
>
> To further clarify the differences,please refer to the table summarized in our response to Q3.
>
> Q7. Is the accumulated inference latency acceptable in long-horizon tasks?
>
> R7. Since consecutive states are often highly similar[1], this
>  issue could  be alleviated by performing planning periodically rather than at every step, reducing accumulated inference cost.
>
> [1].Tang, Yao, et al. "Learning versatile skills with curriculum masking." Advances in Neural Information Processing Systems 37 (2024): 65562-65582.
>
> Q8. Does the current evaluation sufficiently support the claim of generalization beyond the dataset under significant distribution shift?
>
> R8. We report the generalization results in Table 2 of Section 5.3,where environments are initialized with states randomly sampled from the random dataset and evaluated using policies pretrained on other datasets. RAD shows clear improvements over DiffuserLite and other baselines. Appendix D.4 provides additional distribution-shift experiments. To further demonstrate generalization, we conducted a severe state distribution shift evaluation on AntMaze by removing trajectories from one quadrant of the maze and testing from unseen initial states in that region. Detailed setup and results are provided in our response to Reviewer wwsk (R2). These results suggest that RAD is more robust under severe state distribution shifts.

---

> > ### Author Rebuttal · Reviewer_aXSD · 2026-04-02
> >
> > I appreciate the authors’ rebuttal and the additional clarifications. However, I still have several concerns that were not fully resolved.
> >
> > **About R2 and R3.**
> >
> > The construction of the expert database introduces a strong data-selection bias, but the paper does not rule out simpler explanations for the gains. Since target states are retrieved only from a manually defined expert subset, the improvements may come largely from biasing the method toward better data rather than from the proposed retrieval-guided planning mechanism itself. A necessary baseline would be to train BC/IQL/DiffuserLite on the same expert or top-return subset; without such a comparison, it remains unclear whether RAD is benefiting from a stronger method or simply from a stronger data filter.
> >
> > **About R5 and R6.**
> >
> > Although RAD is different from static data augmentation methods such as DiffStitch, I remain unconvinced that it should be viewed as a fully standalone offline RL method. It does not generate a fixed augmented dataset, so it is not a standard augmentation approach in that sense. At the same time, its planning component still depends fundamentally on existing diffusion backbones such as Diffuser and DiffuserLite. This makes RAD feel more like a retrieval-guided extension built on top of prior generative planners than a self-contained new algorithmic framework, which in turn limits the strength of the novelty claim.
> >
> > **About R7.**
> >
> > I do not find the efficiency and practicality claims fully convincing. On AntMaze, DL-RAD increases per-step inference time from 0.06s to 0.26s, which is roughly a 4x slowdown. Even in locomotion, the inference time is approximately doubled, while the reported gain is modest (86.0 vs. 87.5), and Table 3 does not provide statistical significance. More importantly, the paper does not analyze how this additional latency accumulates in long-horizon settings, where per-step overhead can become substantial. The rebuttal mentions periodic planning as a possible mitigation, but this remains a conjecture unless supported by experiments. Therefore, I do not think the current evidence is sufficient to justify the claim that the latency is acceptable for real-time execution.
> >
> > At this stage, these concerns remain significant enough that I will keep my overall recommendation unchanged.
> >
> > ---------
> >
> > I appreciate the authors’ effort during the reply rebuttal comment by authors, which has improved my understanding of the paper and partially addressed some of my concerns.
> >
> > - First, I acknowledge that novelty is partly subjective, and I can accept the authors’ argument to some extent. However, I still view RAD less as a fully standalone offline RL method and more as a plug-and-play retrieval-based enhancement built on top of existing diffusion planning frameworks. In fact, as the authors state, “RAD can be seamlessly integrated into any diffusion-based planning framework, demonstrating its generality and plug-and-play nature rather than any limitation.” While this supports the claimed flexibility of the method, it also reinforces my impression that RAD is better understood as an add-on mechanism than as a self-contained offline RL algorithm. Even if it is not a standard data augmentation method like DiffStitch, this still tempers the strength of the originality claim in my view.
> >
> > - Second, the additional significance test and periodic replanning results are useful, but I remain somewhat unconvinced by the efficiency claim. In particular, the reported full-rollout wall-clock time appears to mix inference cost with rollout length, and the AntMaze variance is extremely large, making the practical efficiency benefit harder to interpret clearly. Thus, although this concern is also partially addressed, I do not think the current evidence is fully convincing.
> >
> > Overall, I appreciate the authors’ clarifications and additional experiments, and they do increase my confidence in the technical soundness of the work. However, I still do not find the contribution strong enough for ICML acceptance. I am therefore willing to raise my score to 3, but not beyond that.

---

> > > ### Author Response · Authors · 2026-04-07
> > >
> > > Q1:Are RAD’s gains mainly due to expert-subset filtering rather than retrieval?
> > >
> > > R1:We train BC / IQL / DiffuserLite on the exact same expert subset used to build RAD’s retrieval database. RAD itself remains unchanged.
> > > | Task | BC (Full) | BC (Exp) | IQL (Full) | IQL (Exp) | DL (Full) | DL (Exp) | RAD |
> > > | - | :-: | :-: | :-: | :-: | :-: | :-: | :-: |
> > > | Antmaze-Medium-Play | 0.0 | 50.0 | 65.8 | 46.0 | 78.0 | 46.7 | 86.7 |
> > > | Antmaze-Large-Play | 0.0 | 13.3 | 42.0 | 30.0 | 72.0 | 40.0 | 80.0 |
> > >
> > > These results show the gains cannot be explained by expert-subset filtering alone.
> > >
> > > Q2:Since RAD depends fundamentally on existing diffusion backbones,does this imply that RAD lacks novelty?
> > >
> > > R2:RAD is not a data augmentation method. Its novelty lies in a retrieval-based mechanism that uses high-return reference trajectories to improve OOD generalization beyond the diffusion backbone itself. This design is backbone-agnostic and can be plugged into any diffusion-based planning framework. We validate RAD on Diffuser and DiffuserLite because they represent two distinct planning paradigms, demonstrating its broad applicability rather than any methodological restriction.
> > >
> > > Q3:Are the locomotion gains in Table 3 significant, and can periodic replanning reduce latency?
> > >
> > > R3:The values 86.0 and 87.5 in Table 3 are averages over 9 locomotion tasks; per-task means and standard errors are reported in Table 1.A paired t-test on the 9 task-level mean scores gives p = 0.041,indicating significance at the 5% level.To address latency, we measure full-rollout wall-clock time with replanning every 10 environment steps:
> > >
> > > | Environment | Replanning interval | Score | Full-rollout inference time (s) | Time reduction vs. 1-step replanning |
> > > | - | :-: | :-: | :-: | :-: |
> > > | Antmaze-Medium-Play| 1 step | 86.7 | 147.1 ± 148.5 | — |
> > > | Antmaze-Medium-Play| 10 steps | 86.7 | 33.6 ± 20.1 | -113.5 s (-77.2%) |
> > > | Hopper-Medium-Replay | 1 step | 100.4 | 109.9 ± 1.1 | — |
> > > | Hopper-Medium-Replay  | 10 steps | 99.8 | 43.2 ± 0.8 | -66.7 s (-60.7%) |
> > >
> > > These results show that periodic replanning substantially reduces RAD’s practical inference latency with almost no performance loss.
> > >
> > > ***
> > > Q1: RAD lacks novelty because it is a plug-and-play enhancement.
> > >
> > > A1: We agree that RAD can be viewed as a plug-and-play enhancement in terms of implementation. However, we respectfully disagree that a plug-and-play method lacks novelty. We believe the community judges novelty by whether a method solves an important problem in a non-trivial way. In fact, some of the most influential works in modern ML are explicitly plug-and-play, such as LoRA (ICLR 2022 Oral, 10k+ citations) and Batch Normalization (ICML 2015, 50k+ citations). RAD uses retrieval to identify high-reward states as planning targets to enhance the decision-making in the OOD area, which shifts diffusion planning from open-loop generation to retrieval-conditioned generation,  addressing the OOD generalization problem in offline RL.
> > >
> > > To further demonstrate that RAD is not the "trivial combination" of retrieval and diffusion.
> > > We further conduct a more challenging quadrant-removal experiment on AntMaze, where trajectories from one quadrant are removed and evaluation is performed from unseen initial states in that region under both 20% and 0% coverage settings. We compare DRC(Diffuser-Lite  naively combined with retrieval) and  DL-RAD (our method). As shown in the table below, DL-RAD consistently outperforms DRC on the more challenging settings. This gap confirms that RAD's gains are not from naive modular stacking.
> > > | Dataset | Coverage Setting |DRC| DL-RAD|
> > > | :--- | :---: | :---: | :---: |
> > > | Antmaze-Medium-Play | $0$% coverage | $20.0$| $33.3$ |
> > > | Antmaze-Medium-Play | $20$% coverage | $26.7$| $52.2$ |
> > > | Antmaze-Large-Play | $0$% coverage | $0.0$| $0.0$ |
> > > | Antmaze-Large-Play | $20$% coverage | $16.7$| $45.6$ |
> > >
> > > Q2: The reported full-rollout wall-clock time appears to conflate inference cost with rollout length and exhibits very large variance on AntMaze.
> > >
> > > A2: To disentangle inference cost from rollout length, we additionally measured the wall-clock time over the same environment steps under different replanning intervals. The results are shown below.
> > > | Dataset | RAD (interval = 1) | RAD (interval = 10) | DL |
> > > | :--- | :---: | :---: | :---: |
> > > | AntMaze-Medium-Play | 37.2±5.5s | 16.0±0.8s | 13.1±0.5s |
> > > | AntMaze-Large-Play | 23.0±0.4s | 9.4±0.2s | 8.5±0.3s |
> > >
> > > Under a fixed rollout horizon, periodic replanning substantially reduces RAD’s inference overhead, bringing its wall-clock time close to DiffuserLite.
> > >
> > > As for the large variance on AntMaze in previous response mainly comes from the benchmark protocol rather than RAD itself: successful episodes often terminate early after reaching the goal, while failures may run the full 1000 steps, naturally leading to high wall-clock variance. This is also seen in the baseline, where DiffuserLite on antmaze-medium-play shows 19.1 ± 11.2s rollout time.

---

### Official Review · Reviewer_fy8w · 2026-03-05

**Soundness:** 3
**Presentation:** 3
**Significance:** 2
**Originality:** 2
**Overall Recommendation:** 4
**Confidence:** 4

**Summary:**

This paper proposes RAD, an offline RL method that introduces a dynamic retrieval mechanism to improve generalization beyond the training distribution. From the authors, RAD retrieves high-return and reachable states from an expert database and uses a diffusion model to generate "recovery" trajectories conditioned on these targets. The method is has two variants, one D-RAD (built on Diffuser) and DL-RAD (built on DiffuserLite). Experiments are conducted on D4RL benchmarks with locomotion, navigation, and manipulation tasks, with marginal/no improvements on D4RL, and only apparent improvement in antmaze.

Overall, I think the paper poses an important issue of generalization for offline RL, but the premise that offline RL fundamentally fails to generalize outside the training distribution is an overstatement, and the proposed solution does not fully address the underlying problem. Specifically, if the offline dataset lacks trajectories connecting the current state to the retrieved target, the diffusion model cannot be expected to generate feasible bridging trajectories, as it is itself trained on the same limited data distribution.

**Compliance With Llm Reviewing Policy:**

Affirmed.

**Final Justification:**

The rebuttal has addressed my concerns. I have raised my score accordingly.

**Key Questions For Authors:**

* Recent work (OGBench, Park et al. 2025) shows offline RL can already stitch trajectories effectively. Under what conditions does RAD's retrieval actually provide meaningful gains over standard offline RL?
* The diffusion model is trained on the same offline dataset. If the dataset lacks coverage between the current state and the retrieved target, why would the diffusion model generate feasible bridging trajectories?
* Why use raw cosine similarity over state vectors rather than a learned representation? How does this hold up in higher-dimensional settings?
* Why is the target selected based on longest remaining trajectory (Eq. 4) rather than highest return? How does this compare against alternative selection criteria?

**Limitations:**

yes

**Strengths And Weaknesses:**

Strengths
* Reliance on dataset coverage is an existing and open challenge in offline RL
* Leveraging a retrieval mechanism to discover reachable high-return states for planning is an interesting idea
* The paper is mostly well-written and easy to follow

Weaknesses
* The premise that offline RL fundamentally fails to generalize outside the dataset is an overstatement. Recent work such as OGBench demonstrates that offline RL can stitch trajectories effectively, and the paper does not engage with this literature or evaluate their methods on the benchmark.
* The retrieval mechanism relies on raw state rather than a learned representation, which may fail to capture semantically meaningful state proximity. This is demonstrated in the fact that the authors use different metrics for different tasks for similarity.
* The criterion for selecting the target state to be the longest remaining trajectory is a heuristic with no theoretical or empirical justification. Why not the trajectory with the highest return?
* The step estimation module's training procedure and necessity are underexplained. It is unclear why temporal alignment cannot be handled implicitly by the diffusion model itself
* The reliance on Diffuser and DiffuserLite for trajectory generation does not resolve the fundamental data coverage problem: if the offline dataset lacks trajectories connecting the current state to the retrieved target, there is no guarantee the diffusion model can generate feasible bridging trajectories, as diffusion models are trained to stay close to the data distribution they were trained on
* The gains are marginal or inconsistent across a large portion of the benchmark: DL-RAD improves only marginally over DiffuserLite on MuJoCo locomotion (87.5 vs. 86.0) and underperforms on Kitchen (72.1 vs. 74.0), suggesting the retrieval mechanism provides genuine benefit only in maze settings.
* Only 3 seed is reported, which is hard to establish statistical significance, particularly in settings where the margins between methods are small
---
References:

Park, S., Frans, K., Eysenbach, B., and Levine, S. OGBench: Benchmarking Offline Goal-Conditioned RL. *In 13th International Conference on Learning Representations (ICLR)*, 2025.

---

> ### Author Rebuttal · Authors · 2026-03-31
>
> Q1. OGBench shows that offline methods can already stitch trajectories effectively. How should RAD be positioned relative to this literature, and under what conditions does retrieval provide meaningful gains over standard offline RL?
>
> R1. The methods provided in OGBench and our RAD differ fundamentally in both experimental settings and methodology.
>
> (1) **Experimental setting**. Existing offline RL methods are usually evaluated when connectable trajectory segments already exist in the dataset, so trajectory stitching is feasible. In contrast, RAD targets a harder setting where such connecting segments may be absent, and the current state can even be out-of-distribution (OOD).
>
> (2) **Methodology**. Prior methods typically rely on **implicit** stitching through value-function generalization (e.g., HIQL), which can be unreliable under highly multimodal behavior. For example, in navigation tasks where the same goal can be reached by going left or right around an obstacle, value functions may average over these modes, producing suboptimal or even infeasible actions. RAD instead performs **explicit** retrieval-based stitching by selecting nearby high-return states as intermediate targets. Since these targets correspond to valid behavior patterns in the dataset, RAD better preserves multimodality and reduces reliance on value extrapolation.
>
> Q2. If the offline dataset lacks coverage between the current state and the retrieved target, why should the diffusion model be able to generate feasible bridging trajectories?
>
> R2. We agree that generating feasible bridging trajectories is challenging for any method. However, diffusion models have shown the ability to learn data manifolds and perform local interpolation beyond memorization[1], and in RAD this capability is further grounded by retrieved target segments, which provide structural constraints that ensure the generated bridging trajectories remain consistent with feasible behaviors in the dataset. We also evaluate a severe distribution-shift setting in AntMaze by removing trajectories from one quadrant and testing from unseen initial states there. As detailed in our response to Reviewer wwsk (R2), RAD still remains effective even in some 0% coverage regions.
>
> [1].Jain, Siddhant. "Video interpolation with diffusion models." Proceedings of the IEEE/CVF Conference on Computer Vision and Pattern Recognition. 2024.
>
> Q3. Why does RAD retrieve neighbors in the raw state space rather than in a learned latent space？
>
> R3. Please see our response to R5 from Reviewer wwsk.
>
> Q4. Does the use of different task-specific similarity metrics make RAD heavily hand-crafted?
>
> R4. Please see our response to R2 from Reviewer HuaD.
>
> Q5. Why is the target selected based on the longest remaining trajectory rather than alternatives such as the highest-return trajectory?
>
> R5. We do not rely on trajectory length alone. We first filter for high-return states, then select the one with the longest remaining trajectory among them. This favors earlier entry into high-value segments rather than isolated late states, providing a more stable anchor for diffusion planning and enough future trajectory support for long-horizon tasks. Comparison with a highest-return-only strategy (Table 1 at https://anonymous.4open.science/r/ICML_data-2EF7/data.pdf) further validates this design.
>
> Q6. Is the step estimation module necessary, and why can temporal alignment not be handled implicitly by the diffusion model?
>
> R6. We believe the SE module is important because the retrieved target specifies where to go, but not when to arrive. The pair $(s_t, s_t^g)$ does not determine the arrival step, so forcing the diffusion model to infer it implicitly makes conditioning less precise and may cause overshooting, stalling, or degenerate trajectories. Our ablations also show that removing SE hurts performance, confirming the value of explicit temporal alignment.
>
> Q7. Why are the empirical gains marginal or inconsistent across several benchmarks, seemingly providing clear benefit mainly in maze settings?
>
> R7. We agree that RAD does not produce uniformly large gains on every benchmark, but we do not think its benefits are limited to maze tasks. RAD also improves locomotion and manipulation tasks, though the gains are smaller because these datasets are richer and already allow strong performance from many methods, including BC, leaving less room for improvement. Moreover, Table 2 shows that RAD’s advantages become more evident under OOD settings, and this trend is not specific to mazes.
>
> Q8. Are 3 random seeds sufficient to support statistical significance, especially when the margins between methods are small?
>
> R8. Thank you for this question. We follow common practice in prior offline RL and diffusion-planning work (e.g., DiffStitch; AdaptDiffuser) and report results over 3 random seeds for fair comparison. Partial results with 5 random seeds are shown in Table 2  (https://anonymous.4open.science/r/ICML_data-2EF7/data.pdf).

---

> > ### Author Rebuttal · Reviewer_fy8w · 2026-04-02
> >
> > R1: The authors refers to this challenging experimental setting where stiching data is not present in the offline dataset but evaluates the method on standard benchmark (D4RL) where prior works can already stitch. Additionally, the authors claim that "Prior methods typically rely on implicit stitching through value-function generalization (e.g., HIQL), which can be unreliable under highly multimodal behavior." without any evidence of this actually happening in their evaluations. This is not a very convincing argument to me.
> >
> > R7 futher confirms the fact that the benchmark that the authors use is saturated and stiching is either achieved or not needed.
> >
> > These concerns remain significant enough. Therefore, I will keep my score unchanged.

---

> > > ### Author Response · Authors · 2026-04-06
> > >
> > > **Q**: Given that D4RL is a standard benchmark where prior works can already achieve trajectory stitching, could the authors clarify why experiments are still conducted on D4RL, rather than evaluating directly on OGBench?
> > >
> > > **R**: Thank you for the comment. First, in this work, we focus on the problem of how an agent can make better decisions to achieve long-term high returns when it encounters out-of-distribution (OOD) states. This objective is slightly different from the stitching capability evaluated in OGBench, which primarily emphasizes explicitly composing trajectories from disconnected sub-trajectories. We do not directly include experiments on OGBench. Instead, we conduct experiments under a **modified D4RL setting (Table 2 and Appendix D.4)**. We train the policy on the standard D4RL datasets (expert, medium, and medium-replay), and evaluate it by initializing from states randomly sampled from the D4RL random dataset.  This setup introduces a clear distribution shift, requiring the agent to generalize beyond the training distribution and make coherent decisions from unfamiliar states.
> > >
> > > To address your concern, we further compare our method with the strongest baseline in OGBench, namely HIQL, under this setting. The results are shown below. As can be seen, RAD consistently outperforms HIQL under the distribution shift case.
> > > |Dataset|HIQL|DiffStitch|DiffuserLite|RAD|
> > > |-|:-:|:-:|:-:|:-:|
> > > |Antmaze-Medium-Play|$37.5$|$36.7$|$6.7$|**$43.3$**|
> > > |Antmaze-Large-Play|$21.9$|$33.3$|$0.0$|**$36.7$**|
> > >
> > > Second, to further demonstrate the effectiveness of our method under distribution shift scenarios, following Reviewer wwsk's suggestion, we additionally performed a more challenging quadrant-removal experiment on AntMaze: we remove trajectories from one quadrant and test from unseen initial states in that region, including 20% coverage and 0% coverage settings. The results are shown below:
> > > | Dataset | Coverage Setting |HIQL| DiffuserLite | RAD |
> > > | :--- | :---: | :---: | :---: |:---: |
> > > | Antmaze-Medium-Play | $0$% coverage | $3.1$| $0.0$ | **$33.3$** |
> > > | Antmaze-Medium-Play | $20$% coverage | $40.6$| $0.0$ | **$52.2$** |
> > > | Antmaze-Large-Play | $0$% coverage | $0.0$| $0.0$ | $0.0$ |
> > > | Antmaze-Large-Play | $20$% coverage | $0.0$| $0.0$ | **$45.6$** |
> > >
> > > RAD remains effective even in some 0% coverage regions. This directly supports our claim that the method remains effective even when the current state is OOD.
> > >
> > > Finally, following your suggestion, we also evaluated RAD on the stitching and explore setting of OGBench. The results are shown below:
> > >
> > > | Dataset | QRL |CRL| HIQL | RAD |
> > > | :--- | :---: | :---: | :---: |:---: |
> > > | Antmaze-Medium-Stitch | $59$  |$53$ | $94$  | **$97$**  |
> > > | Antmaze-Large-Stitch | $18$  | $11$ | $67$  |**$77$**  |
> > > | Antmaze-Medium-Explore | $1$  | $3$ | $37$  |**$97$**  |
> > > | Antmaze-Large-Explore | $0$   | $0$ | $4$  |**$30$**  |
> > >
> > > RAD performs best in both settings. This shows that RAD is effective not only in the more challenging OOD settings emphasized in our paper, but also on standard benchmarks specifically designed for trajectory stitching.
> > >
> > > Finally, we would like to sincerely thank you again for your valuable comment. We will update both the new AntMaze quadrant-removal results and the OGBench results in the revised manuscript.

---

### Official Review · Reviewer_wwsk · 2026-03-10

**Soundness:** 3
**Presentation:** 3
**Significance:** 2
**Originality:** 3
**Overall Recommendation:** 5
**Confidence:** 3

**Summary:**

This paper proposes Retrieval High-quAlity Demonstrations (RAD), a retrieval-augmented planning framework that dynamically selects high-return, reachable states from an offline dataset to serve as intermediate target for a diffusion-based trajectory generator. RAD aims to overcome the limited out-of-distribution generalization and inflexibility of standard offline RL algorithms caused by finite, static datasets. Experimental results demonstrate that RAD delivers superior or competitive performance compared to representative offline RL baselines across diverse continuous control and navigation benchmarks.

**Compliance With Llm Reviewing Policy:**

Affirmed.

**Final Justification:**

Thanks to the authors for the additional experiments about the inference efficiency. This paper is well-structured with clear logic, and the proposed method is novel and interesting. RAD achieves superior performance over the baseline approaches. I have raised my score for supporting the paper.

**Key Questions For Authors:**

1. Could you provide the training cost of RAD compared to baseline methods?

2. What does the -TS mean in ablation study? What is the difference between the experiment provided in D.2?

3. Can RAD maintain its performance under severe state distribution shifts? For example, in the AntMaze environment, if the training data only contains trajectories starting from the top-left and bottom-left reaching a goal in the top-right, how would the policy perform when evaluated from a completely unseen starting position in the bottom-right?

4. The current similarity metric in the TS module relies on basic geometric distance operating on raw state vectors. Have you considered using learned latent embeddings for the similarity search, which maybe better to apply in high-dimensional observation spaces (e.g., images)?

**Limitations:**

yes

**Strengths And Weaknesses:**

### Strengths:

1. This paper proposes a novel and interesting method to expand the state distribution in offline datasets and address the brittleness of static data augmentation methods.

2. The paper is well-structured, and the core methodology is clearly explained. Figure 1 provides a clear intuitive motivation for the approach.

3. This paper conducts comprehensive experiments to verify the effectiveness of each component in RAD and the results demonstrate the superiority of RAD compared to baseline methods.

### Weaknesses:

1. The proposed method incurs a substantial inference overhead, taking approximately four times longer than the baselines. Additionally, the paper lacks clarity regarding the total training cost, which is an important consideration given that the framework optimizes several independent components.

2. It remains unclear whether RAD can maintain its superior performance under severe state distribution shifts, particularly when encountering states that are far removed from the expert trajectories in the offline dataset.

3. There are several inaccuracies in the equations and notations, such as, $\mu_\theta (x_t,t)$ (line 112, also in Eq. 9), and $\phi_\theta$ in Eq. 1, and notation of step estimation model in pseudocode.

---

> ### Author Rebuttal · Authors · 2026-03-31
>
> **Q1.** Could you provide the training cost of RAD compared to baseline methods?
>
> **R1.** Thank you for this suggestion. We provide a more detailed training time which includes backbone training, SE-module training. While the total training time is longer than that of Diffuser and DiffuserLite, we believe it is still acceptable in practice. Moreover, these components can be trained in parallel, which further improves the practical efficiency of our method.
> |Dataset|Backbone Training |SE-module Training |Total training time|
> |-|:-:|:-:|:-:|
> |Navigation |$25$h|$4$h|$29$h|
> |Locomotion |$8.4$h|$3$h|$11.4$h|
>
> **Q2.** Can RAD maintain its superior performance under severe state distribution shifts, especiawhen evaluated on states far from expert trajectories or from unseen starting positions?
>
> **R2.** Yes. To evaluate this, we conduct experiments under severe state distribution shifts using Antmaze-Medium-Play and Antmaze-Large-Play. For each dataset, we first constructed a region-restricted training set by removing all trajectories that enter the bottom-right region, and evaluated both DiffuserLite and RAD from initial states sampled specifically from that unseen region. We further considered a partial-coverage setting, where only 20% of the data in the bottom-right region is retained.
> For fairness, under each setting, both methods were trained on the same training set and evaluated from the same initial states. We report the comparison between DiffuserLite and RAD below.
> | Dataset | Coverage Setting | DiffuserLite | RAD |
> | :--- | :---: | :---: | :---: |
> | Antmaze-Medium-Play | 0\% coverage | $0.0$ | $33.3$ |
> | Antmaze-Medium-Play | 20\% coverage | $0.0$ | $52.2$ |
> | Antmaze-Large-Play | 0\% coverage | $0.0$ | $0.0$ |
> | Antmaze-Large-Play | 20\% coverage | $0.0$ | $45.6$ |
>
> These results suggest that RAD is more robust than DiffuserLite under severe state distribution shifts. On Antmaze-Medium-Play, RAD achieves 33.3 under 0% coverage and 52.2 under 20% coverage, while DiffuserLite fails in both settings. On Antmaze-Large-Play, both methods fail under 0% coverage, but under 20% coverage, RAD reaches 45.6 whereas DiffuserLite remains at 0.0. We attribute RAD’s failure on Antmaze-Large-Play at 0% coverage to the greater long-horizon difficulty of the larger maze: without any bridging support, even retrieval-guided planning cannot reliably connect the held-out region to successful trajectories. Overall, RAD is more effective when even limited retrieval support is available.
>
> **Q3.**  Inaccuracies in equations, notations, and the step estimation model notation in the pseudocode.
>
> **R3.** Thank you for pointing this out. We will correct the affected equations and pseudocode notation in the final version.
>
>
> **Q4.** What exactly does “-TS” mean in the ablation study, and how is it different from the experiment in Appendix D.2?
>
> **R4.** Thank you for the question. In our paper, these two choices are used in two different ablations. In Section 5.4, the “-TS” result is the ablation on D-RAD, where removing TS causes the planning module to fall back to the backbone, so this result should be interpreted as effectively the backbone result. In contrast, Appendix D.2 reports the ablation on DL-RAD, where we retain our PL module and only remove the target-state guidance. Therefore, Appendix D.2 is a within-model ablation, whereas “-TS” in Section 5.4 is a backbone fallback.
>
> **Q5.** Have you considered using learned latent embeddings instead of simple geometric similarity on raw states, especially for high-dimensional observations?
>
> **R5.** We deeply appreciate this insightful suggestion. We agree that learned latent embeddings are a promising alternative to simple geometric similarity, especially for high-dimensional observations. In this work, retrieval in the latent space is primarily treated as a component whose role is to support policy learning. Our focus is on validating the effectiveness of retrieving reachable high-return states, rather than optimizing the retrieval module itself. While improving the accuracy of latent-space retrieval is an important direction, we consider it beyond the scope of the current study. We therefore leave a more thorough investigation of this aspect to future work.

---

> > ### Author Rebuttal · Reviewer_wwsk · 2026-04-01
> >
> > Thank you for your further analysis. My concerns have been addressed and I will keep my score.

---

> > > ### Author Response · Authors · 2026-04-01
> > >
> > > Thank you very much for your thoughtful review and for taking the time to engage with our rebuttal. We are truly encouraged by your response, and we are honored that our clarifications have addressed your concerns. We will carefully incorporate your suggestions into the final version to further improve the quality and clarity of our work.
> > >
> > > ----------------------------------------------------------------------------------------
> > >
> > > **Q1:** The excessive time cost during both training and inference remains a drawback.
> > >
> > > **A1:**
> > > For training, RAD has the same backbone training time as the diffusion-policy baseline. The only additional cost comes from training the SE module. However, this module can be trained in parallel with backbone training, therefore the additional practical overhead is limited. We report the detailed training time below.
> > >
> > > |Dataset|Backbone Training |SE-module Training |
> > > |-|:-:|:-:|
> > > |Navigation |$25$h|$4$h|
> > > |Locomotion |$8.4$h|$3$h|
> > >
> > > For inference, in our original evaluation, target selection was performed after every interaction with the environment, i.e., once a new state was observed, a new target state was retrieved for planning. However, since consecutive states are often highly similar[1], target selection does not necessarily need to be performed at every step. Instead, it can be carried out once every several steps, and the same retrieved target can be reused for planning over the intermediate similar states within that interval. Therefore, this issue can be mitigated by performing planning periodically rather than at every step, thereby reducing the accumulated inference cost. However, periodic planning may also lead to some performance degradation. To verify whether RAD can preserve performance while reducing inference cost, we conducted additional experiments. Specifically, we compared the full-rollout wall-clock time and performance of RAD under two settings: replanning every 10 environment steps and replanning at every step. The results are summarized below.
> > >
> > > | Environment | Replanning interval | Score | Full-rollout inference time (s) | Time reduction vs. 1-step replanning |
> > > | - | :-: | :-: | :-: | :-: |
> > > | Antmaze-Medium-Play| 1 step | 86.7 | 147.1 ± 148.5 | — |
> > > | Antmaze-Medium-Play| 10 steps | 86.7 | 33.6 ± 20.1 | -113.5 s (-77.2%) |
> > > | Hopper-Medium-Replay | 1 step | 100.4 | 109.9 ± 1.1 | — |
> > > | Hopper-Medium-Replay  | 10 steps | 99.8 | 43.2 ± 0.8 | -66.7 s (-60.7%) |
> > >
> > > These results show that periodic replanning substantially reduces the full-rollout inference time while causing only a negligible performance change. On AntMaze-medium-play, increasing the replanning interval from 1 step to 10 steps reduces the full-rollout inference time from 147.1 ± 148.5s to 33.6 ± 20.1s, i.e., a 77.2% reduction, while the score stays the same at 86.7. On Hopper-medium-replay, the full-rollout inference time drops from 109.9 ± 1.1s to 43.2 ± 0.8, i.e., a 60.7% reduction, while the score only decreases slightly from 100.4 to 99.8. Overall, these results provide direct evidence that periodic replanning can substantially mitigate RAD's additional inference latency in practice, with almost no performance degradation.
> > >
> > > [1].Tang, Yao, et al. "Learning versatile skills with curriculum masking." Advances in Neural Information Processing Systems 37 (2024): 65562-65582.
> > >
> > > **Q2:** Could the authors clarify whether the reported full-rollout wall-clock time conflates inference cost with rollout length, and why the variance on AntMaze is so large?
> > >
> > > **A2:** To disentangle inference cost from rollout length, we additionally measured the wall-clock time over the same environment steps under different replanning intervals. The results are shown below.
> > > | Dataset | RAD (interval = 1) | RAD (interval = 10) | DL |
> > > | :--- | :---: | :---: | :---: |
> > > | AntMaze-Medium-Play | 37.2±5.5s | 16.0±0.8s | 13.1±0.5s |
> > > | AntMaze-Large-Play | 23.0±0.4s | 9.4±0.2s | 8.5±0.3s |
> > >
> > > Under a fixed rollout horizon, periodic replanning substantially reduces RAD’s inference overhead, bringing its wall-clock time close to DiffuserLite.
> > >
> > > As for the large variance on AntMaze in previous response mainly comes from the benchmark protocol rather than RAD itself: successful episodes often terminate early after reaching the goal, while failures may run the full 1000 steps, naturally leading to high wall-clock variance. This is also seen in the baseline, where DiffuserLite on antmaze-medium-play shows 19.1 ± 11.2 s rollout time.

---

### Official Review · Reviewer_HuaD · 2026-03-13

**Soundness:** 3
**Presentation:** 2
**Significance:** 3
**Originality:** 3
**Overall Recommendation:** 4
**Confidence:** 3

**Summary:**

The work proposes the Retrieval High-quality Demonstrations (RAD) method to enhance the generalization of offline reinforcement learning in out-of-distribution (OOD) scenarios. This method consists of three parts: the Target Selection module dynamically retrieves high-return, reachable states from the dataset based on similarity, the Step Estimation module predicts the number of steps required to reach these targets, and the Planning module leverages a diffusion-based generative model to synthesize sub-trajectories conditioned on both the current and retrieved states. Experimental results across diverse benchmarks confirm that each component is essential to RAD’s performance.

**Compliance With Llm Reviewing Policy:**

Affirmed.

**Final Justification:**

Thanks to the authors for the response. I am keeping my original rating.

**Key Questions For Authors:**

1. The paper mentioned that RAD relies on the coverage of the offline dataset. In cases where the dataset is purely "medium" quality without any "expert" trajectories, how significantly does the performance of the TS module drop? Would the method benefit from an iterative process where it updates its own database?
2. The current retrieval uses cosine similarity or Euclidean distance. For much higher-dimensional state spaces (e.g., pixel-based inputs), do you anticipate the vector-based similarity search in the TS module becoming a significant bottleneck, and how would you adapt the retrieval mechanism for such cases?
3. The method uses different metrics for different tasks (cosine similarity for locomotion, Euclidean for navigation). Was there a specific heuristic used to make this choice, and is RAD sensitive to using the "wrong" metric for a given task?
4. The planning horizon $H$ varies across environments. Does RAD struggle if the retrieved target state is too far beyond the planning horizon $H$, or does the per-step dynamic retrieval naturally mitigate this by selecting a new target in the next step?

**Limitations:**

yes

**Strengths And Weaknesses:**

## Strengths

1. The work addresses a fundamental bottleneck in offline RL—distributional shift. By shifting from static augmentation to dynamic, per-step planning toward retrieved targets, it offers a more flexible approach to decision-making.



2. The methodology is technically grounded, integrating generative diffusion models with a retrieval mechanism that is theoretically supported by an entropy reduction proof. The authors provide two variants, D-RAD and DL-RAD, demonstrating the framework's compatibility with different diffusion-based backbones.



3. The empirical evaluation is comprehensive, covering diverse tasks (MuJoCo, AntMaze, Kitchen, Maze2D) and comparing against a wide range of baselines, including imitation learning, model-free, model-based, and other data-augmentation methods.

## Weaknesses

1. While the paper claims reachability of target states, the success of the method depends heavily on the coverage of the "expert" database. If the dataset lacks reachable high-return states for a specific OOD region, the planning performance may degrade

2. The retrieval mechanism is not "one-size-fits-all." It requires different similarity metrics for different tasks—cosine similarity for locomotion but Euclidean distance for navigation. This suggests the user must have prior knowledge of the task's geometry for RAD to work effectively.

---

> ### Author Rebuttal · Authors · 2026-03-31
>
> **Q1.** How dependent is RAD on expert-state coverage, especially in medium-only datasets or OOD regions without reachable high-return states? Would iterative database updates help?
>
> **R1.** We partially agree with your point: to some extent, RAD does depend on the coverage of expert states. RAD does indeed fail in OOD regions without reachable high-return states. In such cases, iterative database updates could be useful in the online setting, since the update process may gradually incorporate reachable high-return states. However, in the offline setting, the dataset is static, so iterative database updates are not allowed.
>
> However, RAD is still effective even on medium-only datasets. Specifically, (1) in Table 1, RAD still achieves strong performance on AntMaze; (2) in Table 2, it also performs well on Medium-Replay under distribution shift; and (3) additional severe state-distribution-shift evaluation on AntMaze. Detailed setup and results are provided in our response to Reviewer wwsk (R2).
>
> **Q2.** Do different tasks require different similarity metrics? How sensitive is it to metric choice?
>
> **R2.** We note that a different evaluation metric was adopted in our previous implementation. To ensure consistency and reproducibility, we follow the same metric in this paper. However, further experiments show that the difference between these two metrics is not particularly significant. Specifically, we compare position-only Euclidean distance and full-state cosine similarity on navigation and locomotion tasks.
> The results show that RAD is compatible with multiple metric choices.
>
> |Navigation Task|Cosine|Euclidean|
> |-|:-:|:-:|
> |Antmaze-medium-play|$83.3$|$86.7$|
> |Antmaze-large-play|$76.7$|$80.0$|
>
> |Locomotion Task|Euclidean|Cosine|
> |-|:-:|:-:|
> |HalfCheetah-Medium-Replay|$44.0$|$44.4$|
> |HalfCheetah-Medium|$48.3$|$48.8$|
> |HalfCheetah-Medium-Expert|$88.6$|$90.1$|
>
>
> **Q3.** Can the TS retrieval mechanism scale to high-dimensional inputs, such as pixel-based state spaces, without becoming a bottleneck?
>
> **R3.** For pixel-based observations, RAD would perform retrieval in a learned latent embedding space produced by an encoder as mentioned by reviewer wwsk, which avoids direct nearest-neighbor search over high-dimensional images. Therefore, pixel-based state spaces are not the main bottleneck. The primary focus of this paper is to validate the effectiveness of retrieving reachable high-return states for planning and policy learning. While further optimizing retrieval accuracy remains an interesting direction for future work, it is beyond the current scope of this study.
>
>
> **Q4.** What happens when the retrieved target lies beyond the planning horizon? Does per-step dynamic retrieval mitigate this issue?
>
> **R4.** In our current design, this case is explicitly avoided. In the TS module, after retrieving similar states, we only consider candidate targets within the next $H−1$ steps of each retrieved state. Therefore, the selected target is always an intermediate target within the current planning horizon, rather than an arbitrarily distant final goal.
>  More importantly, retrieval is performed at every decision step. So even if the agent is still far from the final high-value region after executing one action, the next state triggers a new retrieval cycle. In this sense, RAD operates as a receding-horizon guidance mechanism: it repeatedly selects locally reachable high-value waypoints, rather than relying on a single long-range target.

---

> > ### Author Rebuttal · Reviewer_HuaD · 2026-04-01
> >
> > I appreciate the authors' response. My concerns are resolved, and I am keeping my score.

---

> > > ### Author Response · Authors · 2026-04-01
> > >
> > > Thank you very much for your thoughtful review and for taking the time to engage with our rebuttal. We are truly encouraged by your response, and we are honored that our clarifications have addressed your concerns. We will carefully incorporate your suggestions into the final version to further improve the quality and clarity of our work.

---

### Decision · Program_Chairs · 2026-04-30

**Decision:**

Accept (regular)

**Comment:**

This paper proposes RAD, a retrieval-augmented planning framework for offline reinforcement learning that dynamically selects high-return, reachable states from an offline dataset as intermediate planning targets. A diffusion-based generative model then synthesizes sub-trajectories conditioned on these targets, enabling better generalization to out-of-distribution (OOD) states. The framework is instantiated in two variants (D-RAD and DL-RAD) built on Diffuser and DiffuserLite, respectively.

Reviewers broadly agreed that the paper tackles an important challenge in offline RL (distributional shift and OOD generalization) with a well-motivated approach. The paper provides an entropy-reduction theoretical justification and broad empirical evaluation across D4RL benchmarks. The rebuttal substantially strengthened the paper: new quadrant-removal experiments on AntMaze, OGBench evaluations showing RAD outperforms HIQL on both stitching and exploration tasks, ablations against a naive retrieval-diffusion combination (DRC), and periodic replanning experiments demonstrating substantial inference time reduction.

Reviewer aXSD maintained that RAD is better understood as a retrieval-based add-on to existing diffusion planning frameworks rather than a new algorithmic framework for offline RL. The DRC ablation and OGBench results partially mitigate this concern (especially the DRC ablation indicating that the method is not a naive combination). Reviewer fy8w noted that performance gains are marginal or inconsistent on standard D4RL benchmarks (e.g., marginal improvement on locomotion, underperformance on Kitchen). The clearest gains are confined to maze-navigation settings, which may reflect the niche applicability of the retrieval mechanism. The reviewer also pointed out that D4RL is already saturated for trajectory stitching. The OGBench results provided in rebuttal do partially address this, and the OOD/distribution-shift experiments provide meaningful clarification.

The paper is technically sound. In revision, the authors should incorporate all experiments from the rebuttal into the final manuscript, e.g., the OGBench results, quadrant-removal experiments, and DRC ablations. They should clarify efficiency measurements and revise the related work and positioning to better address the OGBench literature and limited gains on non-maze benchmarks. The novelty framing should be tightened to focus on retrieval-conditioned diffusion planning for OOD offline RL.